# Quantitative prediction of rate constants and its application to organic emitters

**Katsuyuki Shizu** [1] **& Hironori Kaji** [1] ✉

Many phenomena in nature consist of multiple elementary processes. If we can predict all the rate constants of respective processes quantitatively, we can comprehensively predict and understand various phenomena. Here, we report that it is possible to quantitatively predict all related rate constants and quantum yields without conducting experiments, using multiple-resonance thermally activated delayed fluorescence (MR–TADF) as an example. MR–TADFs are excellent emitters because of its narrow emission, high luminescence efficiency, and chemical stability, but they have one drawback: slow reverse intersystem crossing (RISC), leading to efficiency roll-off and reduced device lifetime. Here, we show a quantum chemical calculation method for quantitatively obtaining all the rate constants and quantum yields. This study reveals a strategy to improve RISC without compromising other important factors: radiative decay rate constants, photoluminescence quantum yields, and emission linewidths. Our method can be applied in a wide range of research fields, providing comprehensive understanding of the mechanism including the time evolution of excitons.

Multiple-resonance thermally activated delayed fluorescence (MR–TADF) has attracted substantial attention in organic light-emitting diodes (OLEDs) research because it can realise high external quantum efficiency and narrow electroluminescence spectra with high colour purity[1–3]. Hatakeyama et al. reported the first MR–TADF molecule (DABNA-1) in 2016[1] by incorporating B and N atoms into a carbon-based molecular structure. In DABNA-1, the highest occupied molecular orbital (HOMO) and the lowest unoccupied molecular orbital (LUMO) distributions are spatially separated on different atoms, resulting in a relatively small energy difference $\Delta E(T_1 \to S_1)$ of 0.15–0.2 eV between the lowest excited singlet state ($S_1$) and lowest triplet state ($T_1$). The $\Delta E(T_1 \to S_1)$ value is sufficiently small to cause TADF but rather large; the rate constant ($k_{RISC}$) for reverse intersystem crossing (RISC) is small ($9.9 \times 10^3 \, s^{-1}$)[1]. Since the report of DABNA-1, a number of MR–TADF molecules have been developed; however, most of them exhibit $k_{RISC}$ values on the order of $10^4$–$10^5 \, s^{-1\,1-7}$. The slow RISC is considered to cause triplet-related annihilation[8], resulting in efficiency roll-off in OLEDs. The slow RISC also reduces the device lifetime of OLEDs. Thus, there is an urgent need to develop MR–TADF emitters with large $k_{RISC}$ to solve these problems without sacrificing the rate constant of fluorescence from $S_1$ to the ground state ($S_0$) ($k_F(S_1 \to S_0)$ or simply $k_F$), the photoluminescence quantum yield (PLQY), and the colour purity. Recently, Yasuda's group developed MR–TADF materials with $k_{RISC}$ of $10^8 \, s^{-1}$ by utilising the heavy atom effect of selenium (Se) atoms[9]. However, the $k_F$ (-$10^5 \, s^{-1}$) was more than two orders of magnitude smaller than the oxygen (O) type analogue. Yang's group developed Se-containing MR–TADF materials. The $k_{RISC}$ of $10^6 \, s^{-1}$ and the $k_F$ of -$10^7 \, s^{-1}$ are more well-balanced[10]. However, the efficiency roll-off problem has not been well solved; simultaneous realisation of $k_{RISC} > 10^7 \, s^{-1}$ and $k_F > 10^7 \, s^{-1}$ is desirable.

Among all the reported MR–TADF molecules, the RISC mechanisms have been analysed in detail for DABNA-1, DABNA-2, and v-DABNA. Quantum chemical calculations[11–14] and time-resolved photoluminescence (PL) measurements[15] indicate that the total RISC process of the three materials occurs via higher triplet states ($T_n$, $n \geq 2$), typically $T_2$ (Kim et al.[13] carried out an analysis including from $T_1$ to $T_3$), although only the direct $T_1 \to S_1$ RISC has initially been considered. When RISC occurs via $T_2$ ($T_1 \to T_2 \to S_1$), $k_{RISC}$ increases with decreasing $T_2 \to S_1$ (especially when $S_1$ is higher in energy than $T_2$) and $T_1 \to T_2$ energy gaps (denoted as $\Delta E(T_2 \to S_1)$ and $\Delta E(T_1 \to T_2)$, respectively), and

[1]Institute for Chemical Research, Kyoto University, Uji, Kyoto 611-0011, Japan. ✉e-mail: kaji@scl.kyoto-u.ac.jp

increasing $S_1$–$T_2$ spin–orbit coupling (SOC($S_1$–$T_2$)). A small $|\Delta E(T_2 \to S_1)|$ and large SOC($S_1$–$T_2$) accelerate the $T_2 \to S_1$ transition, and a small $\Delta E(T_1 \to T_2)$ accelerates the $T_1 \to T_2$ internal up-conversion.

We have reported three methods to enhance RISC: (1) intervening in a locally excited state between charge transfer type singlet ($^1$CT) and triplet ($^3$CT) states[16], (2) using the fluctuational effect for $^3$CT → $^1$CT RISC[17], although it seemingly violates the El-Sayed rule[18], and (3) using the heavy atom effect[19–21]. Regarding the third method, a practical strategy for increasing $k_{RISC}$ is assumed to incorporate third-, fourth-, or lower-row elements to enhance SOC by their heavy atom effect. Various O-, sulfur- (S-), and Se-containing molecules have been developed for conventional TADF emitters composed of donor and acceptor segments[10,22–30]; $k_{RISC}$ is enhanced by 2–20 times via O → S substitution[10,22,23,25], and up to 50 times via O → Se substitution[10]. The carbonyl (C = O) group can also enhance SOC because of the n–$\pi^*$ orbital; as evidenced by benzophenone, a representative carbonyl compound, having a large rate constant ($k_{ISC}$) for intersystem crossing (ISC) of ~$10^{11}$ s$^{-1}$ [31]. The C = O group is a promising alternative to S and Se for enhancing SOC with only C, H, and O atoms[5,6,32–35]. A C = O-containing MR–TADF emitter developed by the Zysman–Colman group (DDiKTa[6]) exhibited a larger $k_{RISC}$ of $6.3 \times 10^5$ s$^{-1}$ than that of v-DABNA ($k_{RISC}$ = $2.0 \times 10^5$ s$^{-1}$), although DDiKTa had a larger $\Delta E(T_1 \to S_1)$ of 0.16 eV than v-DABNA (0.07 eV), suggesting that the C = O groups in DDiKTa accelerated RISC.

Thus, understanding the excited-state decay mechanisms of MR–TADF emitters is important to design novel materials with enhanced TADF properties. To understand the decay mechanisms of a TADF emitter, it is common to determine the rate constants of electronic transitions from the experimental PLQY and transient photo-luminescence decay curve fitted by a linear combination of exponential decay functions. A comprehensive understanding of the emission mechanism is achieved only when the rate constants for all elementary electronic transitions have been determined. However, it is difficult to determine all rate constants when the number of the rate constants is larger than that of experimentally determined fitting parameters (specifically, when singlet and triplet states energetically higher than $S_1$ and $T_1$ are involved). Our theoretical method allows us to quantitatively predict rate constants and quantum yields, including those inaccessible from experiments. Our method also clarifies the quantitative dynamics (time evolutions) of excitons, providing a comprehensive understanding of the TADF mechanism. Therefore, our method proposed in this study offers a guideline for designing TADF emitters with enhanced properties.

Here, we report a quantitative theoretical investigation of the emission mechanism of MR–TADF molecules, focusing on the acceleration of RISC. As with various phenomena, the emission here consists of multiple elementary processes. The quantitative description of the rate constants not only for RISC but also for all relevant elementary processes is of primary importance because such a description enables a comprehensive understanding and prediction of the emission mechanism. We investigate the RISC mechanism of previously synthesised MR–TADF emitters (BNOO, BNSS, and BNSeSe) (Fig. 1), reported by Yang's group[10]. Here, we define a total ISC/RISC rate constant ($k_{toISC}$/$k_{toRISC}$) as the entire ISC/RISC process involving both the $S_1 \to T_1$/$T_1 \to S_1$ and $S_1 \to T_2 \leftrightarrow T_1$/$T_1 \leftrightarrow T_2 \to S_1$ transitions, which correspond to the experimentally obtained values (see the "Methods" section). Electronic states higher than $S_1$ and $T_2$ are not required to be considered because they are energetically well separated. All the calculated $k_{toISC}$ ($8.6 \times 10^7$, $2.0 \times 10^8$, and $1.0 \times 10^9$ s$^{-1}$ for BNOO, BNSS, and BNSeSe, respectively) and $k_{toRISC}$ ($9.2 \times 10^3$, $2.5 \times 10^5$, and $1.5 \times 10^6$ s$^{-1}$, respectively) in this study well-reproduce the experimental $k_{toISC}$ ($7.5 \times 10^7$, $1.5 \times 10^8$, and $4.9 \times 10^8$ s$^{-1}$, respectively) and $k_{toRISC}$ ($4.3 \times 10^4$, $1.9 \times 10^5$, and $2.0 \times 10^6$ s$^{-1}$, respectively). The calculations also clarify how the S and Se atoms accelerate RISC. All the calculated values of $\Delta E(T_1 \to S_1)$, $k_F(S_1 \to S_0)$, and PLQY ($\Phi$) as well as the prompt and TADF contributions, also

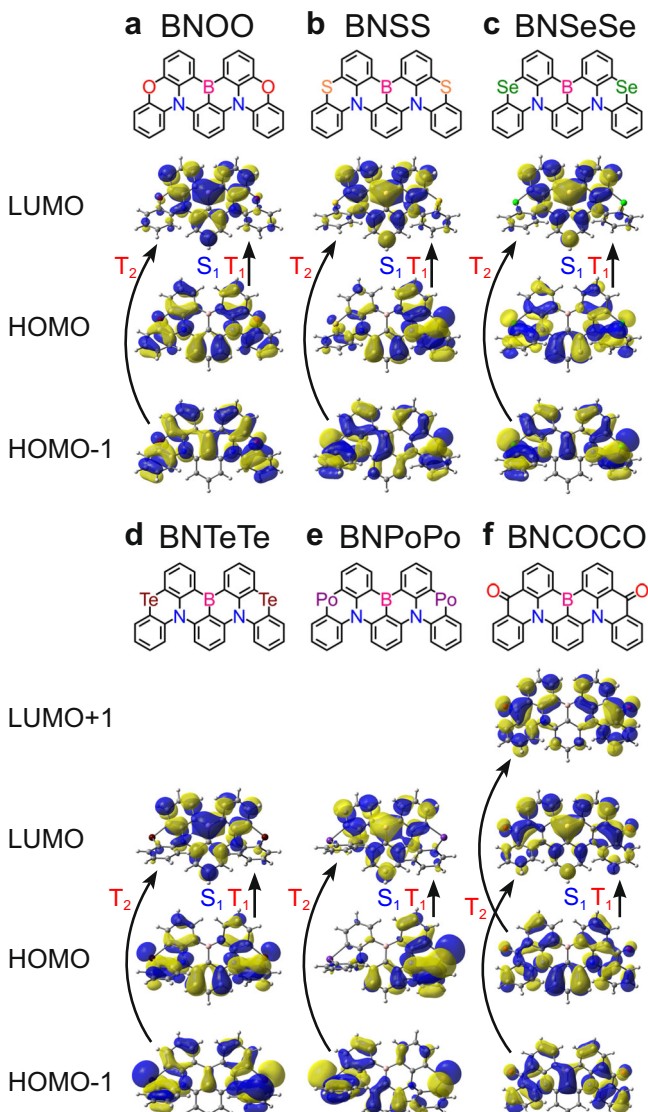

**Fig. 1 | Molecular structures and HOMO−1, HOMO, and LUMO distributions. a** BNOO, **b** BNSS, **c** BNSeSe, **d** BNTeTe, **e** BNPoPo, and **f** BNCOCO. HOMO and LUMO are the highest occupied molecular orbital and lowest unoccupied molecular orbital, respectively.

reasonably reproduce the experimental results (Table 1); indicating the validity of our calculation method. The calculated $k_{toRISC}$ of BNOO slightly deviates from the experimental value because of the over-estimation of $\Delta E(T_1 \to S_1)$ (when we use the experimental $\Delta E(T_1 \to S_1)$ of 0.15 eV instead of the calculated value of 0.21 eV, $k_{toRISC}$ is calculated to be $8.2 \times 10^4$ s$^{-1}$, close to the experimental value of $4.3 \times 10^4$ s$^{-1}$).

Next, we discuss the impacts of further heavy atom effects on RISC and the possibility of further increasing $k_{toRISC}$ by using tellurium (Te), polonium (Po), and C = O substitutions (Fig. 1). Our calculations predict that a Te-containing emitter (BNTeTe) exhibits one order of magnitude larger $k_{toRISC}$ of $10^7$ s$^{-1}$ than BNSeSe ($k_{toRISC}$ ~ $10^6$ s$^{-1}$). Although Po is highly radioactive, we theoretically investigate a Po-containing emitter (BNPoPo) to understand the heavy atom effect. Our calculations indicate that BNPoPo exhibits both TADF and phosphorescence. We calculate the $k_{toRISC}$ of a C = O-containing molecule (BNCOCO) to be $10^6$ s$^{-1}$. Compared with Se, the C = O group has a smaller SOC-enhancement ability but provides a sufficient $T_1 \to T_2$ internal up-conversion, resulting in $k_{toRISC}$ that is comparable to that of BNSeSe. The calculated $k_F(S_1 \to S_0)$s are the same order of magnitude

**Table 1 | Calculated energies, energy gaps, rate constants, quantum yields, spin-orbit couplings, and emission linewidths of BNOO, BNSS, BNSeSe, BNTeTe, BNPoPo, and BNCOCO**

| | BNOO | BNSS | BNSeSe | BNTeTe | BNPoPo | BNCOCO |
|---|---|---|---|---|---|---|
| $E(S_1)$ (eV) | 2.51 (2.54) | 2.44 (2.55) | 2.51 (2.58) | 2.52 | 2.41 | 2.55 |
| $E(T_1)$ (eV) | 2.30 (2.39) | 2.30 (2.42) | 2.37 (2.44) | 2.38 | 2.26 | 2.41 |
| $\Delta E(T_1 \rightarrow S_1)$ (eV) | 0.21 (0.15) | 0.14 (0.13) | 0.14 (0.14) | 0.14 | 0.14 | 0.14 |
| $\Delta E(T_2 \rightarrow S_1)$ (eV) | 0.08 | −0.04 | −0.05 | −0.06 | −0.12 | 0.03 |
| $\Delta E(T_1 \rightarrow T_2)$ (eV) | 0.13 | 0.18 | 0.19 | 0.20 | 0.26 | 0.11 |
| $k_{toISC}$ ($s^{-1}$) | $8.6 \times 10^7$ ($7.5 \times 10^7$) | $2.0 \times 10^8$ ($1.5 \times 10^8$) | $1.0 \times 10^9$ ($0.5 \times 10^9$) | $6.6 \times 10^9$ | $4.3 \times 10^{10}$ | $6.5 \times 10^8$ |
| $k_{toRISC}(T_1)$ ($s^{-1}$) | 8.4 | $2.0 \times 10^3$ | $1.2 \times 10^5$ | $3.0 \times 10^6$ | $4.9 \times 10^7$ | 39 |
| $k_{toRISC}(T_2)$ ($s^{-1}$) | $9.2 \times 10^3$ | $2.5 \times 10^5$ | $1.4 \times 10^6$ | $7.2 \times 10^6$ | $6.9 \times 10^6$ | $9.9 \times 10^5$ |
| $k_{toRISC}$ ($s^{-1}$) | $9.2 \times 10^3$ ($4.3 \times 10^4$) | $2.5 \times 10^5$ ($1.9 \times 10^5$) | $1.5 \times 10^6$ ($2.0 \times 10^6$) | $1.0 \times 10^7$ | $5.6 \times 10^7$ | $9.9 \times 10^5$ |
| $k_{toRISC}'$ ($s^{-1}$) | $9.3 \times 10^3$ | $2.5 \times 10^5$ | $1.5 \times 10^6$ | $1.0 \times 10^7$ | n. d. | $9.9 \times 10^5$ |
| $k_F(S_1 \rightarrow S_0)$ ($s^{-1}$) | $2.7 \times 10^8$ ($8.2 \times 10^7$) | $2.2 \times 10^8$ ($4.5 \times 10^7$) | $2.5 \times 10^8$ ($2.6 \times 10^7$) | $2.4 \times 10^8$ | $1.5 \times 10^8$ | $4.7 \times 10^8$ |
| $k_{NR}(S_1 \rightarrow S_0)$ ($s^{-1}$) | $2.0 \times 10^7$ | $2.1 \times 10^7$ | $1.8 \times 10^7$ | $1.8 \times 10^7$ | $2.0 \times 10^7$ | $1.2 \times 10^7$ |
| $k_{ISC}(S_1 \rightarrow T_1)$ ($s^{-1}$) | $7.9 \times 10^4$ | $1.6 \times 10^6$ | $8.7 \times 10^7$ | $1.9 \times 10^9$ | $3.7 \times 10^{10}$ | $2.6 \times 10^4$ |
| $k_{ISC}(S_1 \rightarrow T_2)$ ($s^{-1}$) | $8.6 \times 10^7$ | $2.0 \times 10^8$ | $9.4 \times 10^8$ | $4.7 \times 10^9$ | $5.3 \times 10^9$ | $6.5 \times 10^8$ |
| $k_{IC}(T_2 \rightarrow T_1)$ ($s^{-1}$) | $9.9 \times 10^{12}$ | $2.4 \times 10^{12}$ | $2.2 \times 10^{12}$ | $1.4 \times 10^{12}$ | $4.2 \times 10^{11}$ | $4.7 \times 10^{12}$ |
| $k_{RISC}(T_2 \rightarrow S_1)$ ($s^{-1}$) | $1.3 \times 10^6$ | $2.6 \times 10^8$ | $1.8 \times 10^9$ | $1.5 \times 10^{10}$ | $1.8 \times 10^{11}$ | $7.8 \times 10^7$ |
| $k_{NR}(T_2 \rightarrow S_0)$ ($s^{-1}$) | 0.68 | 2.7 | $1.9 \times 10^2$ | $2.1 \times 10^3$ | $2.9 \times 10^4$ | 0.66 |
| $k_{Phos}(T_2 \rightarrow S_0)$ ($s^{-1}$) | 92 | $1.5 \times 10^3$ | $9.6 \times 10^3$ | $5.7 \times 10^4$ | $3.4 \times 10^5$ | $3.4 \times 10^3$ |
| $k_{IC}(T_1 \rightarrow T_2)$ ($s^{-1}$) | $7.1 \times 10^{10}$ | $2.4 \times 10^9$ | $1.7 \times 10^9$ | $6.5 \times 10^8$ | $1.6 \times 10^7$ | $6.1 \times 10^{10}$ |
| $k_{RISC}(T_1 \rightarrow S_1)$ ($s^{-1}$) | 8.5 | $2.0 \times 10^3$ | $1.2 \times 10^5$ | $3.0 \times 10^6$ | $4.9 \times 10^7$ | 40 |
| $k_{NR}(T_1 \rightarrow S_0)$ ($s^{-1}$) | 3.8 | 14 | $1.4 \times 10^2$ | $1.1 \times 10^3$ | $3.5 \times 10^4$ | 0.44 |
| $k_{Phos}(T_1 \rightarrow S_0)$ ($s^{-1}$) | 1.6 | 10 | $4.9 \times 10^2$ | $6.6 \times 10^3$ | $9.7 \times 10^4$ | 0.25 |
| $k_{Prompt}$ ($s^{-1}$) | $3.7 \times 10^8$ ($1.9 \times 10^8$) | $4.4 \times 10^8$ ($2.0 \times 10^8$) | $1.3 \times 10^9$ ($0.5 \times 10^9$) | $6.8 \times 10^9$ | $4.1 \times 10^{10}$ | $1.1 \times 10^9$ |
| $k_{Delayed}$ ($s^{-1}$) | $7.1 \times 10^3$ ($26 \times 10^3$) | $1.4 \times 10^5$ ($0.5 \times 10^5$) | $3.1 \times 10^5$ ($1.0 \times 10^5$) | $3.9 \times 10^5$ | $3.5 \times 10^5$ | $4.2 \times 10^5$ |
| $k_{TADF} = k_{toR}(S_1)$ ($s^{-1}$) | $6.6 \times 10^3$ | $1.3 \times 10^5$ | $2.9 \times 10^5$ | $3.6 \times 10^5$ | $1.9 \times 10^5$ | $4.1 \times 10^5$ |
| $k_{toR}(T_1)$ ($s^{-1}$) | 1.6 | 10 | $4.9 \times 10^2$ | $6.6 \times 10^3$ | $9.7 \times 10^4$ | 0.25 |
| $k_{toR}(T_2)$ ($s^{-1}$) | 0.65 | 1.4 | 7.1 | 28 | 13 | 43 |
| $k_{toR}$ ($s^{-1}$) | $6.6 \times 10^3$ | $1.3 \times 10^5$ | $2.9 \times 10^5$ | $3.7 \times 10^5$ | $2.9 \times 10^5$ | $4.1 \times 10^5$ |
| $\Phi$ | 0.93 (0.71) | 0.91 (0.91) | 0.93 (1.0) | 0.93 | 0.83 | 0.97 |
| $\Phi_{NR}(S_1)$ | 0.07 | 0.09 | 0.07 | 0.07 | 0.07 | 0.03 |
| $\Phi_{NR}(T_1)$ | 0.00 | 0.00 | 0.00 | 0.00 | 0.10 | 0.00 |
| $\Phi_{NR}(T_2)$ | 0.00 | 0.00 | 0.00 | 0.00 | 0.00 | 0.00 |
| $\Phi_{Prompt}$ | 0.71 (0.43) | 0.50 (0.23) | 0.19 (0.05) | 0.04 | 0.003 | 0.41 |
| $\Phi_{TADF}$ | 0.22 (0.28) | 0.41 (0.68) | 0.74 (0.95) | 0.87 | 0.55 | 0.56 |
| $\Phi_{Phos}(T_1)$ | 0.00 | 0.00 | 0.00 | 0.02 | 0.28 | 0.00 |
| $\Phi_{Phos}(T_2)$ | 0.00 | 0.00 | 0.00 | 0.00 | 0.00 | 0.00 |
| $S_0$-$T_1$ SOC ($cm^{-1}$) | 2.12 | 4.07 | 13.5 | 37.7 | 202 | 0.76 |
| $S_0$-$T_2$ SOC ($cm^{-1}$) | 0.96 | 1.92 | 16.9 | 56.5 | 204 | 0.97 |
| $S_1$-$T_1$ SOC ($cm^{-1}$) | 0.04 | 0.12 | 0.84 | 3.92 | 17.7 | 0.01 |
| $S_1$-$T_2$ SOC ($cm^{-1}$) | 0.55 | 1.17 | 3.32 | 10.6 | 59.1 | 1.01 |
| FWHM (nm) | 50–53 (51) | 50–53 (53) | 47–50 (47) | 47–50 | 52–56 | 43–46 |

Some values are time-dependent; the values here are those at the equilibrium states, which correspond to the experimentally observable ones. The values in the parentheses are experimental data reported by Hu et al.[10]. $k_{TADF}$ for BNOO/BNSS/BNSeSe/BNTeTe/BNPoPo/BNCOCO is defined in the time domain longer than 100/100/10/ 10/1/10 ns. $k_{toRISC}$ for BNOO/BNSS/BNSeSe/BNTeTe/ BNPoPo/BNCOCO is defined in the time domain longer than 1/10/10/10/1/10 ns (Supplementary Figs. 2 and 3). $k_{toRISC}'$ is the total RISC rate constant calculated by our previously proposed method[20]. n.d. means not determined. $k_{toR}(S_1)$, $k_{toR}(T_1)$, and $k_{toR}(T_2)$ are the contributions from $S_1 \rightarrow S_0$ fluorescence, $T_1 \rightarrow S_0$ phosphorescence, and $T_2 \rightarrow S_0$ phosphorescence to $k_{toR}$, respectively: $k_{toR} = k_{toR}(S_1) + k_{toR}(T_1) + k_{toR}(T_2)$; $k_{toR}(S_1) = k_F(S_1 \rightarrow S_0) \times [S_1]/([S_1] + [T_1] + [T_2])$; $k_{toR}(T_1) = k_{Phos}(T_1 \rightarrow S_0) \times [T_1]/([S_1] + [T_1] + [T_2])$; $k_{toR}(T_2) = k_{Phos}(T_2 \rightarrow S_0) \times [T_2]/([S_1] + [T_1] + [T_2])$. $k_{toRISC}(T_1)$ and $k_{toRISC}(T_2)$ are the contributions from $T_1 \rightarrow S_1$ and $T_2 \rightarrow S_1$ ISCs to $k_{toRISC}$, respectively. $k_{toRISC} = k_{toRISC}(T_1) + k_{toRISC}(T_2)$; $k_{toRISC}(T_1) = k_{RISC}(T_1 \rightarrow S_1) \times [T_1]/([T_1] + [T_2])$; $k_{toRISC}(T_2) = k_{RISC}(T_2 \rightarrow S_1) \times [T_2]/([T_1] + [T_2])$.

$(10^8 \text{ s}^{-1})$ for all the compounds investigated in this study. Their calculated PLQYs remain high, indicating that we can enhance RISC without sacrificing radiative decays nor PLQYs, which is otherwise a trade-off in TADF emitters. Finally, we theoretically predict the linewidths (full width at half maximum (FWHM)) of the PL spectra for these molecules. The calculated FWHM values are almost identical for BNOO, BNSS, and BNSeSe; agreeing well with the experimental values. Those for designed BNTeTe, BNPoPo, and BNCOCO are comparable to those for BNOO, BNSS, and BNSeSe; indicating that the acceleration of $k_{RISC}$ in this study also does not sacrifice the FWHM values.

## Results

### Computational results for BNOO, BNSS, and BNSeSe

Our theoretical analysis is based on rate constant calculations. Details of the method of calculating the rate constants for fluorescence from $S_1$ to $S_0$ ($k_F(S_1 \rightarrow S_0)$), $T_n \rightarrow S_0$ phosphorescence ($k_{Phos}(T_n \rightarrow S_0)$), $S_1 \rightarrow S_0$ nonradiative decay ($k_{NR}(S_1 \rightarrow S_0)$), $T_n \rightarrow S_0$ nonradiative decay ($k_{NR}(T_n \rightarrow S_0)$), $T_m \rightarrow T_n$ internal conversion ($k_{IC}(T_m \rightarrow T_n)$), $S_1 \rightarrow T_n$ ISC ($k_{ISC}(S_1 \rightarrow T_n)$), and $T_n \rightarrow S_1$ RISC ($k_{RISC}(T_n \rightarrow S_1)$) ($m, n = 1, 2, m \neq n$) are described in the "Methods" section. As is well known, conventional time-dependent density functional theory (TD-DFT) methods substantially overestimate $\Delta E(T_1 \rightarrow S_1)$ of MR–TADF emitters, which can be solved by calculations including double-excitation configurations[36]. Several wave-function-based methods (including the second-order algebraic-diagrammatic construction ADC(2) and the spin-component-scaling second-order approximate coupled-cluster (SCS–CC2)) give more reliable $\Delta E(T_1 \rightarrow S_1)$ than TD–DFT methods[36]. Recently, it has become common to use different theoretical methods for different molecular properties[13,37,38]. For example, Lin et al. calculated excitation energies with the TD-B3LYP/6-31G(d) method, and then, the $T_1$ energy was corrected using $\Delta E(T_1 \rightarrow S_1)$ obtained from the SCS–CC2/def2-TZVP calculation[38]. Tamm–Dancoff approximation (TDA)–DFT methods with double hybrid density functionals such as TDA–B2-PLYP are emerging alternative approaches for considering double-excitation configurations[39]. TDA–DFT methods have the advantage of low computational cost compared with ADC(2) and SCS–CC2. Here, we compared the $\Delta E(T_1 \rightarrow S_1)$ values calculated with the three methods as well as that by the conventional TD–DFT (B3LYP) method (blue texts in Supplementary Tables 13 and 14). Supplementary Table 13 also shows the experimental values. The TDA–B2-PLYP method provided the $\Delta E(T_1 \rightarrow S_1)$ values closest to the experimental results for BNSS and BNSeSe. Therefore, we used the TDA–B2-PLYP method (TDA–DFT with the B2-PLYP double-hybrid functional and def2-TZVP basis set) for calculating $\Delta E(T_1 \rightarrow S_1)$, which we combined with the energy levels of $S_1$ and $T_2$ calculated by the TD–DFT with the B3LYP functional and 6-31 G(d)+SDD basis set (TD–B3LYP method). All the other calculations (SOCs, vibronic coupling constants, transition dipole moments, and permanent dipole moments) were carried out by the TD–B3LYP method (see Methods section and Supplementary Method 2 for the details). We performed the TDA–B2-PLYP calculations with the ORCA 5.0.3 program package (FACCTs, Cologne, Germany)[40–42] and the ADC(2) and SCS–CC2 calculations with the TURBOMOLE program package[43]. Table 1 shows the calculated and experimental data. Figure 2a–f shows the calculated excited-state energy diagrams, energy gaps, SOCs, and rate constants for BNOO, BNSS, and BNSeSe. Figure 2g–l shows the time evolutions of respective rate constants. The calculated $\Delta E(T_1 \rightarrow S_1)$ for BNSS and BNSeSe agree with the experimental values (calculated and experimental $\Delta E(T_1 \rightarrow S_1)$ are 0.14 and 0.13 eV, respectively, for BNSS and they are both 0.14 eV for BNSeSe), although we found a slight deviation by 0.06 eV for BNOO (the $\Delta E(T_2 \rightarrow S_1)$ and $\Delta E(T_2 \rightarrow T_1)$ values have not been determined experimentally). We can reasonably neglect the contributions of $S_n$ ($n \geq 2$) and $T_m$ ($m \geq 3$) as described above.

For BNOO, BNSS, and BNSeSe, $S_1$ and $T_2$ are energetically close ($|\Delta E(T_2 \rightarrow S_1)| < 0.1$ eV), and the $S_1$–$T_1$ SOCs are stronger than the $S_1$–$T_1$ SOCs (Table 1 and Fig. 2a–c). As a result, $k_{ISC}(S_1 \rightarrow T_2)$ is larger than

$k_{ISC}(S_1 \rightarrow T_1)$ by 1000 times for BNOO, 100 times for BNSS, and 10 times for BNSeSe. $S_1$ and $T_1$ of BNOO, BNSS, and BNSeSe are predominantly described as the HOMO−LUMO transitions, whilst their $T_2$s are predominantly described as the HOMO-1-LUMO transitions (Fig. 1). The $S_1$–$T_2$ SOCs are more substantial than the $S_1$–$T_1$ SOCs because $S_1$–$T_2$ transitions are between different molecular orbital (MO) characters, whilst $S_1$–$T_1$ transitions are between similar MOs.

The larger $S_1$–$T_2$ SOCs compared with $S_1$–$T_1$ SOCs, as well as closer energy levels of $S_1$–$T_2$ compared with those of $S_1$–$T_1$, suggest faster transitions for $T_2$-mediated ISC and RISC compared with those for direct $S_1$–$T_1$ ISC and RISC. The calculated $k_{toISC}$ agrees well with the experimental values for BNOO, BNSS, and BNSeSe (Table 1). Regarding BNOO and BNSS, the stepwise $S_1 \rightarrow T_2 \rightarrow T_1$ transition contributed to the entire ISC process and the direct $S_1 \rightarrow T_1$ ISC was negligible ($k_{toISC} \approx k_{ISC}(S_1 \rightarrow T_2) \gg k_{ISC}(S_1 \rightarrow T_1)$; $k_{IC}(T_2 \rightarrow T_1)$ is very large). Regarding BNSeSe, although the contribution from the direct $S_1 \rightarrow T_1$ ISC was not negligible, the stepwise $S_1 \rightarrow T_2 \rightarrow T_1$ transition was still dominant ($k_{toISC} \approx k_{ISC}(S_1 \rightarrow T_2)$ and $k_{ISC}(S_1 \rightarrow T_1)$ was one order smaller than $k_{ISC}(S_1 \rightarrow T_2)$). $k_{toISC}$ increased with increasing atomic number for the chalcogen atoms ($8.6 \times 10^7 \text{ s}^{-1} < 2.0 \times 10^8 \text{ s}^{-1} < 1.0 \times 10^9 \text{ s}^{-1}$ for BNOO, BNSS, and BNSeSe, respectively), indicating that the O → S → Se substitution enhanced the $S_1$–$T_2$ SOC and accelerated the $S_1 \rightarrow T_2$ ISC.

Next, we investigated RISC for BNOO, BNSS, and BNSeSe. Table 1 shows the calculated and experimental $k_{toRISC}$ as well as $\Phi$, $\Phi_{Prompt}$, $\Phi_{TADF}$, $\Phi_{Phos}(T_1)$, $\Phi_{Phos}(T_2)$, and $k_{TADF}$ for BNOO, BNSS, and BNSeSe. Here, $\Phi_{Prompt}$, $\Phi_{TADF}$, $\Phi_{Phos}(T_1)$, and $\Phi_{Phos}(T_2)$ are the PLQYs of the prompt fluorescence, TADF, phosphorescence from $T_1$, and phosphorescence from $T_2$, respectively. $k_{TADF}$ is the rate constant of TADF (delayed fluorescence). Because $\Phi_{Phos}(T_1) \approx 0$ for the three compounds, we attributed the delayed luminescence to TADF ($\Phi_{Phos}(T_2) \approx 0$ for all six compounds). Most importantly, the calculated $k_{toRISC}$ increased with increasing atomic number for the chalcogen atoms, and the $k_{toRISC}$ quantitatively agrees with the experimental values, which confirms the validity of our method of predicting $k_{toRISC}$. The calculations here enabled us to separate the contributions of direct RISC ($T_1 \rightarrow S_1$) and RISC via $T_2$ ($T_1 \rightarrow T_2 \rightarrow S_1$). We denote them as $k_{toRISC}(T_1)$ ($=k_{RISC}(T_1 \rightarrow S_1)$) and $k_{toRISC}(T_2)$, respectively. Regarding BNOO, BNSS, and BNSeSe, the rate constants for $k_{toRISC}(T_2)$ are almost identical to $k_{toRISC}$ ($=k_{toRISC}(T_1) + k_{toRISC}(T_2)$) (Table 1 and Fig. 2j–l), indicating that the $T_1 \rightarrow T_2 \rightarrow S_1$ transition is the dominant RISC pathway. It should also be noted that $k_{toRISC}$ is much smaller than $k_{RISC}(T_2 \rightarrow S_1)$. This is because the downhill $T_2 \rightarrow T_1$ IC is rapid compared with the uphill $T_1 \rightarrow T_2$ IC and $T_2 \rightarrow S_1$ RISC. Thus, in addition to increasing $k_{RISC}(T_2 \rightarrow S_1)$, decreasing $\Delta E(T_1 \rightarrow T_2)$ is also crucial for accelerating $T_2$-mediated RISC.

### Computational results for BNTeTe and BNPoPo

From the above discussion, we expected further enhanced RISC by further increasing the atomic number of the included atoms. Hence, we next replaced Se with Te or Po (Fig. 1d and e) to further increase the SOCs and accelerate RISC. HOMO − 1, HOMO, and LUMO of BNTeTe and BNPoPo are similar to those of BNOO, BNSS, and BNSeSe (Fig. 1). Regarding BNTeTe (Fig. 3a), the energy level alignment of $S_1$, $T_1$, and $T_2$ is almost identical to those of BNSS and BNSeSe (Fig. 2b, c). Hence, the S/Se→Te replacement affected $k_{toRISC}$ by enhancing the $S_1$–$T_1$ and $S_1$–$T_2$ SOCs. We calculated the $S_0$–$T_1$, $S_1$–$T_1$, and $S_1$–$T_2$ SOCs of BNTeTe to be 37.7, 3.92, and 10.6 cm$^{-1}$, respectively, which are 3 to 4 times those of BNSeSe (13.5, 0.84, and 3.32 cm$^{-1}$, respectively). As a result, all SOC-related rate constants ($k_{ISC}(S_1 \rightarrow T_1)$, $k_{ISC}(S_1 \rightarrow T_2)$, $k_{RISC}(T_2 \rightarrow S_1)$, $k_{RISC}(T_1 \rightarrow S_1)$, $k_{NR}(T_1 \rightarrow S_0)$, and $k_{Phos}(T_1 \rightarrow S_0)$) of BNTeTe were larger than those of BNSeSe. In contrast, $k_F(S_1 \rightarrow S_0)$ and $k_{NR}(S_1 \rightarrow S_0)$ of BNTeTe were comparable to those of BNSS and BNSeSe. Thus, acceleration of RISC is possible without sacrificing radiative decay and PLQY (Fig. 3d, g, j). The contributions of direct ISC and direct RISC to total ISC and total RISC processes, respectively, were more substantial for BNTeTe than for BNSeSe. However, the stepwise $T_2$-mediated

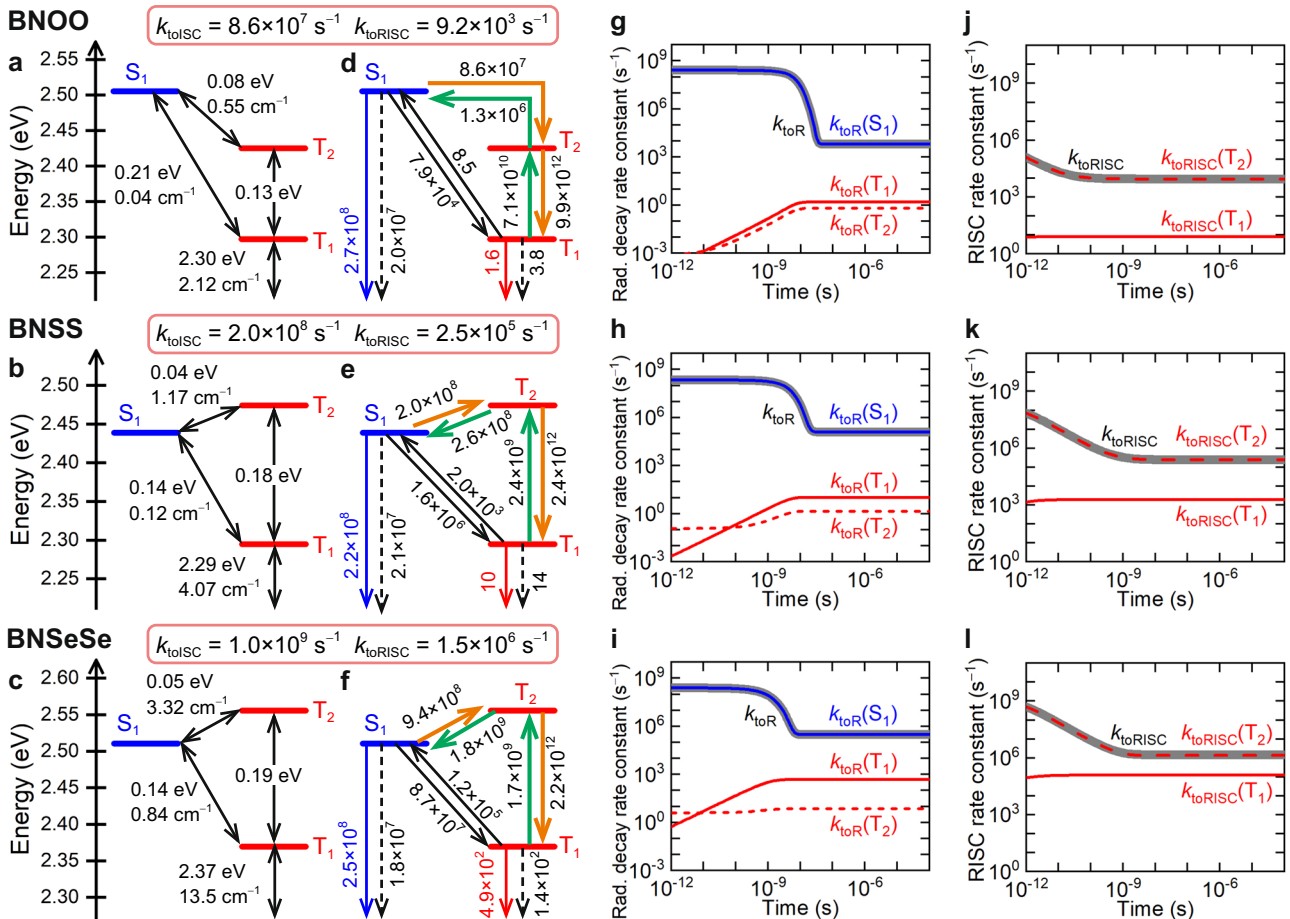

**Fig. 2 | Excited-state decay mechanism. a–c** Calculated excited-state energy diagram, energy differences (eV), and spin–orbit couplings (cm$^{-1}$) and **d–f** rate constants (s$^{-1}$) for BNOO, BNSS, and BNSeSe. **g–i** Calculated $k_{toR}$, $k_{toR}(S_1)$, $k_{toR}(T_1)$, and $k_{toR}(T_2)$ for BNOO, BNSS, and BNSeSe. **j–l** Calculated $k_{toRISC}$, $k_{toRISC}(T_1)$, and $k_{toRISC}(T_2)$ for BNOO, BNSS, and BNSeSe. In **d–f**, the solid orange arrows depict the dominant $S_1 \rightarrow T_1$ pathway, the solid green arrows depict the dominant $T_1 \rightarrow S_1$ pathway, the solid black arrows depict the minor $S_1 \rightarrow T_1$ and $T_1 \rightarrow S_1$ pathways, the

solid blue arrows depict the $S_1 \rightarrow S_0$ fluorescence, the solid red arrows depict the $T_1 \rightarrow S_0$ phosphorescence, and the dashed arrows depict $S_1 \rightarrow S_0$ and $T_1 \rightarrow S_0$ non-radiative decays. In **g–i**, the solid grey curves depict $k_{toR}$, the blue curves depict $k_{toR}(S_1)$, the solid red curves depict $k_{toR}(T_1)$, and the dashed red curves depict $k_{toR}(T_2)$. In **j–l**, the solid grey curves depict $k_{toRISC}$, the solid red curves depict $k_{toRISC}(T_1)$, and the dashed red curves depict $k_{toRISC}(T_2)$. Source data for figure g-l are provided as a Source Data file.

process was still dominant (70% of the total process; Fig. 3j). The calculated $k_{toRISC}$ of BNTeTe was on the order of $10^7$ s$^{-1}$. $k_{Phos}$ also increased but on the order of $10^3$ s$^{-1}$; therefore, phosphorescence was still negligible ($\Phi_{Phos}(T_1) = 0.02$), and $T_1$ excitons were preferentially converted into light as TADF (Fig. 3g). Thus, BNTeTe is a promising MR–TADF emitter with fast RISC.

Po has a more substantial heavy atom effect than Te (Fig. 3b); hence, one can expect that BNPoPo would also be an excellent TADF emitter with much faster RISC (although Po compounds are radioactive, we performed calculations to understand the heavy atom effect). In contrast to our expectation, BNPoPo exhibited substantial phosphorescence as well as TADF ($\Phi_{Phos} = 0.28$ and $\Phi_{TADF} = 0.55$; Table 1 and Fig. 3e show rate constants). Therefore, BNPoPo would not be a pure TADF emitter (Fig. 3h), although it exhibited the fastest $k_{toRISC}$ of $5.6 \times 10^7$ s$^{-1}$ because of the substantial heavy atom effect of Po. This is different from all the other compounds in this study (they are pure TADF emitters). Interestingly, the dominant RISC processes are found to change from $T_1 \rightarrow T_2 \rightarrow S_1$ to direct $T_1 \rightarrow S_1$ process at $10^{-10}$ s for BNPoPo (Fig. 3k). Our proposed method also provides such detailed exciton dynamics.

## Computational results for BNCOCO

We investigated another approach to accelerating RISC: C = O substitution. Figure 1f shows the designed molecule (BNCOCO). We

calculated $\Delta E(T_1 \rightarrow S_1)$ of BNCOCO to be 0.14 eV (Fig. 3c); which is comparable to those of BNSS, BNSeSe, BNTeTe, and BNPoPo. $S_1$–$T_2$ SOC of BNCOCO (1.01 cm$^{-1}$, Fig. 3c) is only slightly smaller than that of BNSS (1.17 cm$^{-1}$, Fig. 2b). Although $T_2 \rightarrow S_1$ RISC is downhill for BNSS and uphill for BNCOCO, the energy gaps for $T_2 \rightarrow S_1$ were small in both cases. Consequently, the $k_{RISC}(T_2 \rightarrow S_1)$ of BNCOCO is only slightly smaller than that of BNSS. In contrast, the reduced $\Delta E(T_1 \rightarrow T_2)$ induced a faster $T_1 \rightarrow T_2$ transition than in BNSS, resulting in a larger $k_{toRISC}$ ($9.9 \times 10^5$ s$^{-1}$) than BNSS ($2.5 \times 10^5$ s$^{-1}$). $k_{toISC}$ and $k_{toRISC}$ of BNCOCO are close to those of BNSeSe. The large spatial overlap between HOMO and LUMO + 1 around the C = O groups (Fig. 1f) lowers the $T_2$ energy level, minimising $\Delta E(T_1 \rightarrow T_2)$ and leading to the large $k_{toRISC}$ even without chalcogen atoms.

## A method for minimising $\Delta E(T_1 \rightarrow T_2)$

Here, we would like to discuss another approach to minimise $\Delta E(T_1 \rightarrow T_2)$, which is effective in increasing $k_{toRISC}$. Supplementary Fig. 8a shows a case when $T_1$ and $T_2$ consist of HOMO → LUMO and HOMO−1 → LUMO transitions, respectively. $\Delta E(T_1 \rightarrow T_2)$ can be written as $\Delta E(T_1 \rightarrow T_2) = (h_{HH} - h_{H-1H-1}) + (J_{HH} - J_{H-1H-1}) + (J_{HL} - J_{H-1L})$, where $h$ denotes the core integral (kinetic and potential energies), $J$ denotes the Coulomb integral, and $K$ denotes the exchange integral[44] (see Supplementary Method 6 for the detail). A simple approach of minimising $\Delta E(T_1 \rightarrow T_2)$ is to decrease the first term $h_{HH} - h_{H-1H-1}$, which is possible

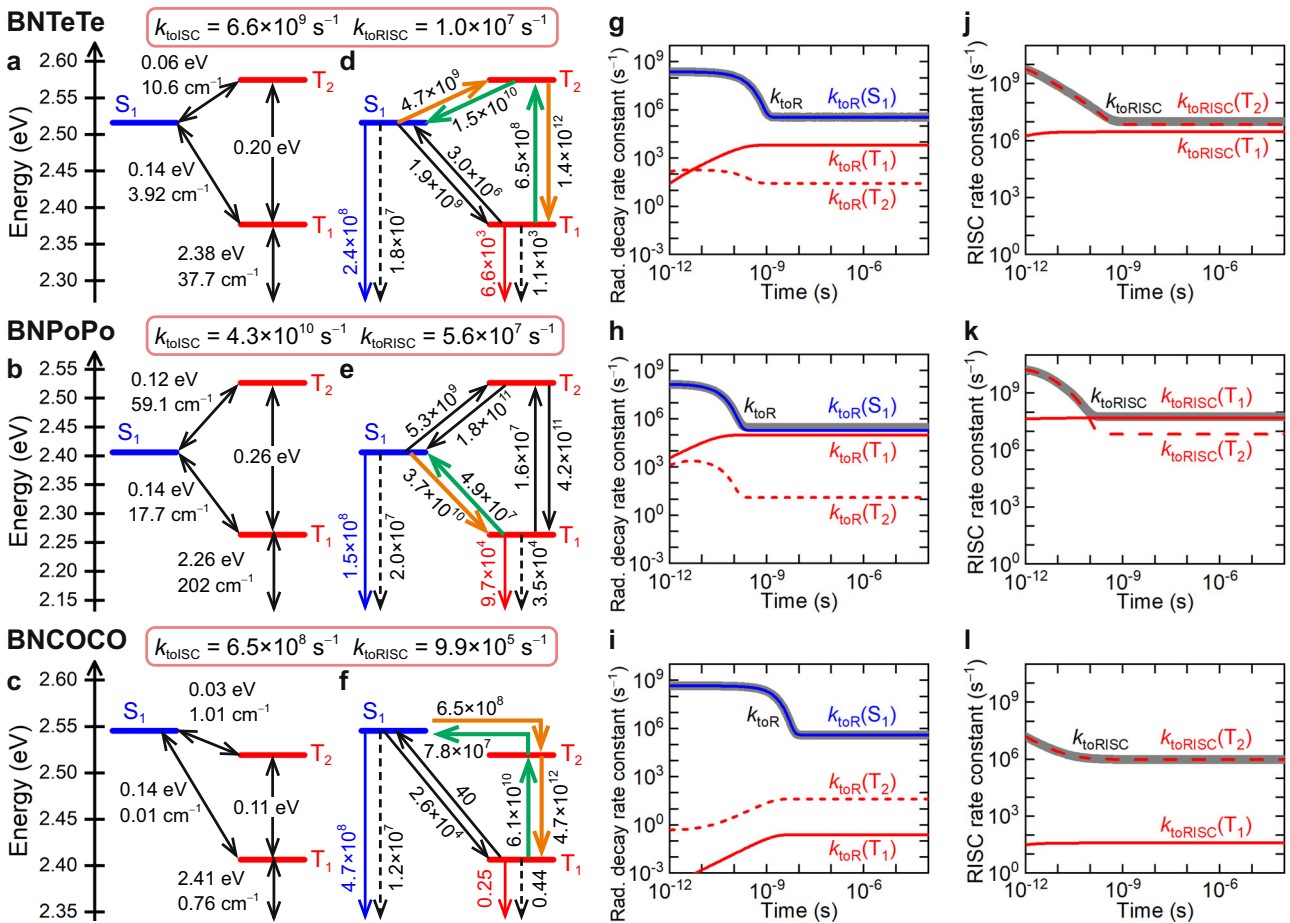

**Fig. 3 | Excited-state decay mechanism. a–c** Calculated excited-state energy diagram, energy differences (eV), and spin-orbit couplings (cm$^{-1}$) and **d–f** rate constants (s$^{-1}$) for BNTeTe, BNPoPo, and BNCOCO. **g–i** Calculated $k_{toR}$, $k_{toR}(S_1)$, $k_{toR}(T_1)$, and $k_{toR}(T_2)$ for BNTeTe, BNPoPo, and BNCOCO. **j–l** Calculated $k_{toRISC}$, $k_{toRISC}(T_1)$, and $k_{toRISC}(T_2)$ for BNTeTe, BNPoPo, and BNCOCO. In **d–f**, the solid orange arrows depict the dominant $S_1 \rightarrow T_1$ pathway, the solid green arrows depict the dominant $T_1 \rightarrow S_1$ pathway, the solid black arrows depict the minor $S_1 \rightarrow T_1$ and $T_1 \rightarrow S_1$

pathways, the solid blue arrows depict the $S_1 \rightarrow S_0$ fluorescence, the solid red arrows depict the $T_1 \rightarrow S_0$ phosphorescence, and the dashed arrows show $S_1 \rightarrow S_0$ and $T_1 \rightarrow S_0$ nonradiative decays. In **g–i**, the solid grey curves depict $k_{toR}$, the blue curves depict $k_{toR}(S_1)$, the solid red curves depict $k_{toR}(T_1)$, and the dashed red curves depict $k_{toR}(T_2)$. In **j–l**, the solid grey curves depict $k_{toRISC}$, the solid red curves depict $k_{toRISC}(T_1)$, and the dashed red curves depict $k_{toRISC}(T_2)$. Source data for **g–l** are provided as a Source Data file.

by expanding the HOMO and HOMO−1 distributions (the second and third terms of $\Delta E(T_1 \rightarrow T_2)$, expressed in terms of the Coulomb integrals, are not easy to control). Supplementary Fig. 8b shows a case where $T_1$ and $T_2$ consist of the HOMO → LUMO and HOMO → LUMO + 1 transitions, respectively. In this case, $\Delta E(T_1 \rightarrow T_2)$ can be written as $\Delta E(T_1 \rightarrow T_2) = (h_{L+1L+1} - h_{LL}) + (J_{HH+1} - J_{HL}) + (K_{HL+1} - K_{HL})$. For the same reason, expanding the LUMO + 1 and LUMO distributions results in an effective approach to minimise $\Delta E(T_1 \rightarrow T_2)$ by decreasing $h_{L+1L+1} - h_{LL}$. Regardless of whether $T_2$ is described as the HOMO−1 → LUMO or HOMO → LUMO + 1 transition, expanding molecular orbitals relevant for $T_1$ and $T_2$ is a simple way to decrease $\Delta E(T_1 \rightarrow T_2)$ and accelerate the $T_2$-mediated RISC process. Comparison of v-DABNA-core and V-DABNA-core is a good example[45]. V-DABNA has a larger π-conjugation than v-DABNA and hence, V-DABNA shows a smaller $\Delta E(T_1 \rightarrow T_2)$ of 98 meV than v-DABNA-core (147 meV). This trend can be also seen in several other examples[46–50] (Supplementary Table 15).

## Exciton dynamics
The above discussion indicates that the rate constants, including those of ISC and RISC, and quantum yields can be predicted quantitatively. This method also provides quantitative details on the exciton dynamics. Figures 2g–l and 3g–l show the time evolutions of respective rate constants. Supplementary Figs. 2 and 3 also show the time evolutions of respective exciton concentrations. These analyses enable

complete elucidation of the emission mechanism. As shown in Figs. 2g–l and 3g–l, the total radiative, ISC, and RISC rate constants, corresponding to experimentally observed ones, are not constant and time-dependent because they are composed of multiple processes. Initially, $k_{toR}$ is very large ($10^8$–$10^9$ s$^{-1}$) but decreases to $10^4$–$10^5$ s$^{-1}$. These time evolutions, corresponding to prompt and delayed emissions, respectively, are automatically calculated by our method, including the transition states. Figures 2j–l and 3j–l show that $k_{toRISC}$'s are also time-dependent; initially, they are very large and stay constant after -$10^{-10}$–$10^{-9}$ s. Figure 3k shows that the RISC mechanism of BNPoPo changed from the $T_2$-mediated RISC to direct $T_1 \rightarrow S_1$ RISC at -0.1 ns as described above.

## Theoretically prediction of linewidths of PL spectra
Finally, we theoretically calculated and predicted the FWHM values of the emission spectra by the vertical gradient method implemented in the ORCA 5.0.3 program package[40–42]. The calculated FWHM values of BNOO, BNSS, and BNSeSe agree well with the experimental values (Table 1 and Supplementary Figs. 4–6), confirming the validity of the calculations. Regarding BNTeTe, BNPoPo, and BNCOCO, the predicted FWHM values were 47–50, 52–56, and 43–46 nm, respectively (Table 1 and Supplementary Fig. 7); which are comparable to those for BNOO, BNSS, and BNSeSe. Thus, acceleration of RISC is possible without sacrificing the FWHM of the emission spectra.

## Discussion

We comprehensively investigated the RISC mechanism of MR−TADF emitters, BNOO, BNSS, and BNSeSe by calculating all the relevant rate constants and quantum yields. The calculated values are in quantitative agreement with the experimental results. The ISC and RISC in BNOO, BNSS, and BNSeSe were found to occur predominantly via $T_2$. We also found that incorporating S and Se into the molecules was effective in accelerating RISC. Therefore, BNTeTe and BNPoPo were designed to further increase $k_{RISC}$. The strong heavy atom effect of Te enabled BNTeTe to exhibit $k_{RISC}$ of $10^7 \, s^{-1}$, which is larger than those of BNOO ($10^4 \, s^{-1}$), BNSS ($10^5 \, s^{-1}$), and BNSeSe ($10^6 \, s^{-1}$). Meanwhile, although BNPoPo had the largest $k_{RISC}$ of $5.6 \times 10^7 \, s^{-1}$, it exhibited a substantial contribution of phosphorescence because of the excessive heavy atom effect of Po. Our findings suggest that a moderately strong SOC that does not cause phosphorescence and suitable energy alignments are the keys to accelerating RISC in pure TADF. We also investigated BNCOCO to attain large $k_{RISC}$ without heavy atoms. BNCOCO had an $S_1-T_2$ SOC comparable to BNSS but a smaller $T_1-T_2$ energy difference, resulting in a larger $k_{RISC}$ of $10^6 \, s^{-1}$.

Distinct from conventional TADF, RISC has been rate-limiting in MR-TADF, and addressing this problem is currently the most important issue in further improving its characteristics. The calculated $k_F(S_1 \rightarrow S_0)$, $\Phi$, and FWHM values of the emission spectra are nearly identical for all the compounds in this study. This study reveals the solution that the rate constants of RISC can be substantially improved without sacrificing the PLQY, rate constant of radiative decay, and emission linewidth.

Finally, we would like to discuss further acceleration of RISC. The $k_{RISC}(T_2 \rightarrow S_1)$ of BNTeTe was large ($1.5 \times 10^{10} \, s^{-1}$) because of the large SOC and downhill transition. However, the $k_{toRISC}$ was three orders of magnitude smaller ($1.0 \times 10^7 \, s^{-1}$). This is because in the dominant process of RISC ($T_1 \rightarrow T_2 \rightarrow S_1$), $k_{IC}(T_2 \rightarrow T_1)$ ($1.4 \times 10^{12} \, s^{-1}$) is much larger than $k_{IC}(T_1 \rightarrow T_2)$ ($6.5 \times 10^8 \, s^{-1}$) and $k_{RISC}(T_2 \rightarrow S_1)$. The small $k_{IC}(T_1 \rightarrow T_2)$ results in a slow pump-up of excitons from $T_1$ to $T_2$. Even if excitons are up-converted from $T_1$ to $T_2$, they quickly return to $T_1$ ([$T_2$] ≪ [$T_1$]). This problem can be solved by minimising the energy difference between $T_1$ and $T_2$ (eventually to zero) yet preserving the $T_2 \rightarrow S_1$ transition downhill. This situation corresponds to a system in which $S_1$ is energetically lower than $T_1$. Recently, it has been revealed that molecules with an $S_1$ that is lower in energy than $T_1$ can be realised despite violating Hund's rule[51–56]. Such inverted $S_1-T_1$ (iST) systems have an ideal energy level diagram that enables the aforementioned downhill direct RISC process without the $T_1 \rightarrow T_n$ IC process. iST molecules have frontier orbital distributions that are similar to MR-TADF; HOMO and LUMO are localised on different atoms, having short-range CT characters. Our calculation method for MR−TADF can be directly applied to designing iST molecules, enabling further enhanced RISC and a comprehensive understanding of their emission mechanisms.

In this study, we proposed a theoretical method of predicting the energy level alignments, including higher lying states as well as $S_1$ and $T_1$, by combining conventional B3LYP and B2-PLYP double-hybrid functionals. Second, we devised a method of calculating RISC (and ISC) rate constants considering RISC (and ISC) mechanisms consisting of multiple pathways. Although only $S_1$, $T_1$, and $T_2$ are involved in the emission mechanism of molecules in this study, the method proposed here is robust enough to be applied to the case where many higher-lying states are involved. The theoretical advances enable us to quantitatively predict all the relevant rate constants and quantum yields. It is demonstrated that it is important to consider the entire system for quantitative understanding of the actual emission mechanism. To quantitatively understand RISC for all the compounds here, it is not sufficient to consider only the $T_1 \rightarrow S_1$ process; IC plays an important role, albeit indirect.

Our method of calculating rate constants and quantum yields is not limited to such MR-TADFs and iSTs. The range of applications is vast, including electronic transitions in, e.g. biochemical, biomedical, pharmaceutical systems, chemical reactions, and surface science. We have also confirmed the applicability of this method to a wide range of TADF emitters other than MR-type molecules and to catalytic photo-oxygenation to inhibit aggregation of amyloid-β peptide as a therapeutic strategy for Alzheimer's disease[57].

One phenomenon often consists of combinations of several elementary processes. Of these processes, only a key process has so far been focused on. However, as in the present example, all processes are often closely related. Our physics-based method is therefore important for obtaining a comprehensive understanding of phenomena and for quantitative predictions, including the time evolutions, which enable discovery of superior systems.

## Methods

### Calculations of rate constants for TADF, phosphorescence, total ISC, total RISC, and radiative decay based on excited-state populations

We calculated the rate constants for fluorescence from $S_1$ to $S_0$ ($k_F(S_1 \rightarrow S_0)$), $T_n \rightarrow S_0$ phosphorescence ($k_{Phos}(T_n \rightarrow S_0)$), $S_1 \rightarrow S_0$ nonradiative decay ($k_{NR}(S_1 \rightarrow S_0)$), $T_n \rightarrow S_0$ nonradiative decay ($k_{NR}(T_n \rightarrow S_0)$), $T_m \rightarrow T_n$ internal conversion ($k_{IC}(T_m \rightarrow T_n)$), $S_1 \rightarrow T_n$ ISC ($k_{ISC}(S_1 \rightarrow T_n)$), and $T_n \rightarrow S_1$ RISC ($k_{RISC}(T_n \rightarrow S_1)$) ($m, n = 1, 2, m \neq n$) with Supplementary Eqs. (A1)–(A18) (Supplementary Method 1). $S_n$ ($n \geq 2$) and $T_m$ ($m \geq 3$) were located 0.3 eV higher in energy than $S_1$, resulting in very small contributions for all compounds in this study; therefore, we neglected their contributions to the TADF mechanism. We performed geometric optimisation and frequency analysis of $S_1$ for BNOO, BNSS, BNSeSe, BNTeTe, BNPoPo, and BNCOCO by the TD-TPSSh method (Supplementary Tables 1−6 and Supplementary Fig. 1 shows the optimised geometries). Then, we performed the excited-state calculations with the TD−B3LYP method using the optimised $S_1$ geometries (Supplementary Tables 7−12). For H, B, C, N, O, and S atoms, we used the 6–31G(d) basis set. For Se, Te, and Po atoms, we used the Stuttgart/Dresden pseudopotentials and basis set (SDD)[58]. The geometrical optimisations, frequency analyses, and excited-state calculations were performed with the Gaussian 16 program package (Wallingford, CT, USA)[59].

Calculations of SOCs, vibronic coupling constants, transition dipole moments, and permanent dipole moments were carried out by the TD−B3LYP method (TD−DFT with the B3LYP functional and 6–31G(d)+SDD basis set), in which the SOCs, vibronic coupling constants, and $T_1-T_2$ transition dipole moments were calculated with the method proposed by McMurchie and Davidson[60] (Supplementary Method 2 and Code availability below), whereas the permanent dipole moments and $S_0-S_1$ transition dipole moment were calculated with the Gaussian 16 program package. We performed the TDA−B2-PLYP calculations with the ORCA 5.0.3 program package (FACCTs, Cologne, Germany)[40–42] and the ADC(2) and SCS−CC2 calculations with the TURBOMOLE program package[43].

The derivations of rate constants of individual elementary processes are shown in Supplementary Method 3. Here, we show the derivations of rate constants composed of multiple processes. Previously, rate constants for prompt and delayed fluorescence and RISC have been determined from transient photoluminescence (trPL) decay curves (number of photons counted vs. time plot)[20], which are difficult to use for separately analysing delayed fluorescence and phosphorescence, especially when their time scales are close. In this study, we propose a method of calculating these rate constants from the excited-state populations, [$S_n$] and [$T_n$] ($n = 1, 2, 3, ...$). Our method distinguishes delayed fluorescence and phosphorescence, which does not require calculating a trPL decay curve.

The rate for the total ISC from the excited singlet states ($S_n$ ($n = 1$, 2, 3, ...)) to the triplet states ($T_m$ ($m = 1, 2, 3, ...$)) can be expressed as

$$\sum_{n \geq 1} \left( \sum_{m \geq 1} k_{ISC}(S_n \to T_m) \right) [S_n], \tag{1}$$

which can be written in terms of the total population of the excited singlet states $\sum_{l \geq 1}[S_l]$ as

$$\left\{ \sum_{n \geq 1} \left( \sum_{m \geq 1} k_{ISC}(S_n \to T_m) \right) \frac{[S_n]}{\sum_{l \geq 1}[S_l]} \right\} \sum_{l \geq 1}[S_l]. \tag{2}$$

Therefore, the rate constant for the total ISC $k_{toISC}$ can be defined as

$$k_{toISC} = \sum_{n \geq 1} \left( \sum_{m \geq 1} k_{ISC}(S_n \to T_m) \right) \frac{[S_n]}{\sum_{l \geq 1}[S_l]}. \tag{3}$$

Here, the effect of $k_{IC}(S_{n'} \to S_{n''})$ on $k_{toISC}$ are included in the $[S_n]$ populations because $[S_n]$ are calculated by solving the kinetic equations that include all the elementary transitions (see Supplementary Method 3).

Furthermore, the rate for the total RISC from the triplet states to the excited singlet states can be expressed as

$$\sum_{m \geq 1} \left( \sum_{n \geq 1} k_{RISC}(T_m \to S_n) \right) [T_m], \tag{4}$$

which can be written in terms of the total population of the triplet states $\sum_{l \geq 1}[T_l]$ as

$$\left\{ \sum_{m \geq 1} \left( \sum_{n \geq 1} k_{RISC}(T_m \to S_n) \right) \frac{[T_m]}{\sum_{l \geq 1}[T_l]} \right\} \sum_{l \geq 1}[T_l]. \tag{5}$$

Therefore, the rate constant for the total RISC $k_{toRISC}$ can be defined as

$$k_{toRISC} = \sum_{m \geq 1} \left( \sum_{n \geq 1} k_{RISC}(T_m \to S_n) \right) \frac{[T_m]}{\sum_{l \geq 1}[T_l]}. \tag{6}$$

Note that $k_{toISC}$ and $k_{toRISC}$ are functions of time ($t$) through the time dependence of $[S_n]$ and $[T_m]$, respectively. As in the case of $k_{IC}(S_{n'} \to S_{n''})$, the effect of $k_{IC}(T_{m'} \to T_{m''})$ on $k_{toRISC}$ are included in the $[T_m]$ populations (see Supplementary Method 3). The values of $k_{toISC}$ and $k_{toRISC}$ depend on the time domain in which they are calculated. The rate for the total radiative decay from the excited singlet and triplet states is

$$\sum_{n \geq 1} k_F(S_n \to S_0)[S_n] + \sum_{m \geq 1} k_{Phos}(T_m \to S_0)[T_m], \tag{7}$$

which can be written in terms of the total population of the excited states $\sum_{l \geq 1}[S_l] + \sum_{l \geq 1}[T_l]$ as

$$\left\{ \sum_{n \geq 1} k_F(S_n \to S_0) \frac{[S_n]}{\sum_{l \geq 1}[S_l] + \sum_{l \geq 1}[T_l]} \right.$$
$$+ \sum_{m \geq 1} k_{Phos}(T_m \to S_0) \frac{[T_m]}{\sum_{l \geq 1}[S_l] + \sum_{l \geq 1}[T_l]} \right\} \tag{8}$$
$$\times \left( \sum_{l \geq 1}[S_l] + \sum_{l \geq 1}[T_l] \right).$$

Hence, the rate constant for the total radiative decay ($k_{toR}$) can be defined as

$$k_{toR} = \sum_{n \geq 1} k_F(S_n \to S_0) \frac{[S_n]}{\sum_{l \geq 1}[S_l] + \sum_{l \geq 1}[T_l]}$$
$$+ \sum_{m \geq 1} k_{Phos}(T_m \to S_0) \frac{[T_m]}{\sum_{l \geq 1}[S_l] + \sum_{l \geq 1}[T_l]}. \tag{9}$$

As in $k_{toISC}$ and $k_{toRISC}$, $k_{toR}$ is a function of $t$.

As stated previously, regarding BNOO, BNSS, BNSeSe, BNTeTe, BNPoPo, and BNCOCO, it is sufficient to consider only $S_1$, $T_1$, and $T_2$. Hence, $k_{toISC}$, $k_{toRISC}$, and $k_{toR}$ can be written as

$$k_{toISC} = k_{ISC}(S_1 \to T_1) + k_{ISC}(S_1 \to T_2) \tag{10}$$

$$k_{toRISC} = k_{RISC}(T_1 \to S_1)\frac{[T_1]}{[T_1]+[T_2]} + k_{RISC}(T_2 \to S_1)\frac{[T_2]}{[T_1]+[T_2]} \tag{11}$$

$$k_{toR} = k_F(S_1 \to S_0)\frac{[S_1]}{[S_1]+[T_1]+[T_2]} + k_{Phos}(T_1 \to S_0)\frac{[T_1]}{[S_1]+[T_1]+[T_2]}$$
$$+ k_{Phos}(T_2 \to S_0)\frac{[T_2]}{[S_1]+[T_1]+[T_2]}. \tag{12}$$

The contributions from $T_1 \to S_1$ and $T_2 \to S_1$ RISCs to $k_{toRISC}$ are defined as

$$k_{toRISC}(T_1) = k_{RISC}(T_1 \to S_1)\frac{[T_1]}{[T_1]+[T_2]}, \tag{13}$$

$$k_{toRISC}(T_2) = k_{RISC}(T_2 \to S_1)\frac{[T_2]}{[T_1]+[T_2]}. \tag{14}$$

The contributions from $S_1 \to S_0$ fluorescence $k_{toR}(S_1)$, $T_1 \to S_0$ phosphorescence $k_{toR}(T_1)$, and $T_2 \to S_0$ phosphorescence $k_{toR}(T_2)$ to $k_{toR}$, are defined as

$$k_{toR}(S_1) = k_F(S_1 \to S_0)\frac{[S_1]}{[S_1]+[T_1]+[T_2]}, \tag{15}$$

$$k_{toR}(T_1) = k_{Phos}(T_1 \to S_0)\frac{[T_1]}{[S_1]+[T_1]+[T_2]}, \tag{16}$$

$$k_{toR}(T_2) = k_{Phos}(T_2 \to S_0)\frac{[T_2]}{[S_1]+[T_1]+[T_2]}. \tag{17}$$

Regarding fluorescent molecules with negligibly small $k_{Phos}(T_1 \to S_0)$ and $k_{Phos}(T_2 \to S_0)$, such as BNOO/BNSS/BNSeSe/BNTeTe/BNCOCO, $k_{toR} \sim k_F(S_1 \to S_0)$ when $[T_1] \ll [S_1]$ and $[T_2] \ll [S_1]$ (this condition holds in the time domain immediately after the $S_0 \to S_1$ photoexcitation). After sufficient time has passed following the excitation, $S_1$, $T_1$, and $T_2$ are thermally equilibrated, and $[S_1]/([S_1] + [T_1] + [T_2])$ becomes constant (Figs. 2 and 3, and Supplementary Figs. 2 and 3). In such a time domain, $k_{toR} \sim k_F(S_1 \to S_0) \times [S_1]/([S_1] + [T_1] + [T_2])$, which can be viewed as the rate constant for TADF ($k_{toR} \sim k_{TADF}$). For molecules that emit both fluorescence and phosphorescence, such as BNPoPo, $k_{toR}$ is intrinsically the population-weighted average of $k_F(S_1 \to S_0)$, $k_{Phos}(T_1 \to S_0)$, and $k_{Phos}(T_2 \to S_0)$ (Eq. (12)). The lifetimes for the total radiative decay ($\tau_{toR}$) and TADF ($\tau_{TADF}$) can be calculated as $\tau_{toR} = 1/k_{toR}$ and $\tau_{TADF} = 1/k_{TADF}$, respectively. Yersin et al. derived $\tau_{toR}$ for organometal complexes[61]. Equations (9) and (12) are corrected from their equations (Supplementary Method 3 shows a detailed comparison).

**Calculations of rate constants for prompt fluorescence, TADF, and RISC based on transient photoluminescence decay curves**

Experimentally, TADF properties have often been discussed in terms of the RISC rate constant ($k_{\text{toRISC}}'$) determined from a trPL decay curve. Here, $k_{\text{toRISC}}'$ denotes the trPL-based rate constant, whilst $k_{\text{toRISC}}$ denotes the excited-state population-based rate constant (Eqs. (6) or (11)). We previously proposed an expression for $k_{\text{toRISC}}'$ under the assumption that the triplet states do not decay radiatively nor nonradiatively[20]:

$$
k_{\text{toRISC}}' = \frac{k_{\text{Prompt}} + k_{\text{Delayed}}}{2} \\
- \sqrt{\left(\frac{k_{\text{Prompt}} + k_{\text{Delayed}}}{2}\right)^2 - k_{\text{Prompt}} k_{\text{Delayed}}\left(1 + \frac{\Phi_{\text{Delayed}}}{\Phi_{\text{Prompt}}}\right)}, \quad (18)
$$

where $k_{\text{Prompt}}$ and $k_{\text{Delayed}}$ denote the rate constants for prompt and delayed luminescence, respectively, determined by exponential fitting of a trPL decay curve. $K_{\text{toRISC}}'$ cannot be applied to emitters having phosphorescent contributions such as BNPoPo. Although $k_{\text{toRISC}}$ is almost identical to $k_{\text{toRISC}}'$ for BNOO, BNSS, BNSeSe, BNTeTe, and BNCOCO (in which $k_{\text{Phos}}(T_1 \rightarrow S_0)$ and $k_{\text{NR}}(T_1 \rightarrow S_0)$ are negligibly small (Table 1)), $k_{\text{toRISC}}$ is applicable to molecules that exhibit prompt fluorescence, delayed fluorescence, and phosphorescence simultaneously; which is more universal than $k_{\text{toRISC}}'$.

## Data availability
The Gaussian 16 and ORCA input and output files are deposited in the figshare data repository [https://doi.org/10.6084/m9.figshare]. The source data underlying Figs. 2g–l and 3g–l are provided as a Source Data file. Source data are provided with this paper.

## Code availability
The code used to generate Table 1 and Figs. 2 and 3 is available on GitHub [https://github.com/KatsuyukiShizu/d77] and Zenodo [https://doi.org/10.5281/zenodo.11124543].

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

## Acknowledgements

The quantum chemical calculations were performed on the Super-Computer System, Institute for Chemical Research, Kyoto University. This work was supported by JSPS KAKENHI grant numbers: JP20H05840 (Grant-in-Aid for Transformative Research Areas, "Dynamic Exciton", H.K.), JP22K05252 (K.S.), and JSPS Core-to-Core Programme: JPJSCCA20220004 (H.K.). We thank Michael Scott Long, PhD, at Edanz (https://jp.edanz.com/ac) for editing a draft of this manuscript.

## Author contributions

K.S. performed the theoretical calculations. H.K. planned and supervised the project. All authors contributed to the writing of this manuscript and have approved the final version.

## Competing interests

The authors declare no competing interests.
