## [Peer Review File · Nature Communications]

Quantitative prediction of rate constants and its application to organic emittersREVIEWER COMMENTS

Reviewer #1 (Remarks to the Author):

Title: Quantitative Prediction of Rate Constants and Quantum Yields in Multiple-Resonance Thermally Activated Delayed Fluorescence (MR-TADF)

The manuscript introduces a novel approach to quantitatively predict rate constants and quantum yields for the process of multiple-resonance thermally activated delayed fluorescence (MR-TADF). The authors highlight the potential of this method for enhancing the performance of organic light-emitting diodes (OLEDs), focusing on MR-TADF emitters and addressing the challenge of slow reverse intersystem crossing (RISC).

Overall, the manuscript presents a well-structured and scientifically sound research study that offers valuable insights into the quantitative prediction of rate constants and quantum yields, particularly in the context of MR-TADF. The study is significant in its potential to improve the efficiency and device lifetime of OLEDs, making it highly relevant to the field.

The key aspect of this paper is the introduction of quantum chemical method for quantitatively reproducing experimentally obtained rate constants and quantum yields, providing a novel and promising approach to predict these parameters without the need for experiments. While this will be highly relevant in the field - its focus on MR-TADF is relatively weak. The examples given address only a few examples - most of whom have limited practical use due to the rare elements included.

The errors compared to experiments are not huge making the results impressive - but I would like to see a more comprehensive analysis of its performance across other examples.

Reviewer #2 (Remarks to the Author):

The manuscript by Shizu and Kaji presents a remarkable method for predicting rate constants of excited-state decay processes of MR-TADF emitters. The relevance of MR-TADF in OLEDs is obvious, but its performance problem due to slow RISC is a major concern. The authors address this issue by quantum chemical calculations to quantitatively predict the rate constants and resulting quantum yields for MR-TADF emitters. The versatility of their approach, with potential

applications in various research areas, adds significant value to the study. I would recommend publication in Nature Communications after addressing the following concerns.

1. The title "Quantitative prediction of rate constants and its application to organic emitters" may sound misleading, since the calculated values are the rate constants of excited-state decays in a single molecule. The same is true of their claim in the abstract.
2. While this manuscript delves into the various practices of MR-TADF emitters, a clearer presentation of the theoretical advances and the reason why they enable the quantitative predictions would greatly aid the reader's understanding.
3. The manuscript does not specify the method used to calculate the SOCs, vibronic coupling constants, and T1-T2 transition dipole moments (it just says that it was done by their own method). Providing this information is essential for the reproducibility of their study.
4. The authors deliberately employed the TDA-B2-PLYP/def2-TZVP to calculate ΔE_{ST} and the TD-B3LYP/6-31G(d)+SDD for the other parameters in predicting the rate constants. The manuscript could benefit from a justification for mixing different methods to ensure that the accuracy of their predictions is not primarily due to error compensation (error canceling).

Reviewer #3 (Remarks to the Author):

In this manuscript, Kaji and coworkers theoretically predicted the rate constants of each process involved in the multi-resonance thermally activated delayed fluorescence (MR-TADF) of a series of MR-TADF emitters consisting of different heavy atoms by quantum chemical calculations. They comprehensively studied the effects of heavy atoms on the kinetics of MR-TADF and the photophysics of the emitters. They showed a clear picture on the kinetics of each elementary process, and the contribution to the total rate constants by the S1, T1 and T2 states was analysed extensively. However, there are numerous studies involving the calculations of these elementary processes, for example, the rate constant calculations using MOMAP by Zhigang Shuai's research group, and it is predictable that the emitter involving more heavy polonium may have phosphorescence even without calculations. Therefore, together with the following comments, this work is not novel enough to be published on Nat. Comm.

1. In this work, the authors have not discussed the influences of heavy atoms on the stability of the MR-TADF emitters, which is critical to the lifetime of the devices. In the devices, there will be some host-guest interactions between the emitter and the host, and so the MR-TADF compounds may

undergo dissociation to form radicals. In addition, the poor operation lifetime of OLED was confirmed to arise from the weak bond energy between the carbon and heavy atoms (Adv. Funct. Mater. 2023, 2306880). The strategy of introducing heavy atoms to the MR–TADF molecules is meaningless if the stability of the MR–TADF emitters is low, even the RISC is accelerated without sacrificing radiative decay and PLQY.

2. On page 5, line 86, the authors quote the experimental k_{RISC} values from Yang's work (ref. 22), which are 4.3×10^4 and $1.9 \times 10^5 \text{ s}^{-1}$, but in ref. 22, the k_{RISC} values provided in Table 1 are not consistent with them.
3. On page 5, line 84, the calculated k_{RISC} of BNSS is 27 times of that of BNOO, but in ref. 22, it is around 2-3 times. How do the authors explain such great discrepancy? The statement of "indicating the validity of our calculation method" is thus inappropriate.
4. On page 6, line 104, how to predict the FWHM of the PL spectra theoretically?
5. On page 6, line 109-126, it is better to include it in the section of computational details instead in discussion section. The same suggestion on lines 143-151 on page 7.
6. On page 11, line 240, the authors state that "The large spatial overlap between HOMO and LUMO+1 around the C=O groups lowers the T2 energy level, leading to the large k_{toRISC} .", The lowering of the T2 energy would increase the barrier for the T2→S1 transition. Why does it lead to large k_{toRISC} ?
7. In supplementary Figure 4, does the simulation of the calculated emission spectra involve the broadening by any functions like Gaussian, Lorentzian and Voigt functions?

Response to the comments of reviewer #1

Reviewer's comment:

Title: Quantitative Prediction of Rate Constants and Quantum Yields in Multiple-Resonance Thermally Activated Delayed Fluorescence (MR-TADF) The manuscript introduces a novel approach to quantitatively predict rate constants and quantum yields for the process of multiple-resonance thermally activated delayed fluorescence (MR-TADF). The authors highlight the potential of this method for enhancing the performance of organic light-emitting diodes (OLEDs), focusing on MR-TADF emitters and addressing the challenge of slow reverse intersystem crossing (RISC).

We would thank the reviewer for the valuable comments. We have revised the manuscript according to your comments as follows. Important changes are highlighted by yellow in the revised manuscript.

Reviewer's comment 1:

Overall, the manuscript presents a well-structured and scientifically sound research study that offers valuable insights into the quantitative prediction of rate constants and quantum yields, particularly in the context of MR-TADF. The study is significant in its potential to improve the efficiency and device lifetime of OLEDs, making it highly relevant to the field.

We appreciate the reviewer's positive comment on our manuscript.

Reviewer's comment 2:

The key aspect of this paper is the introduction of quantum chemical method for quantitatively reproducing experimentally obtained rate constants and quantum yields, providing a novel and promising approach to predict these parameters without the need for experiments. While this will be highly relevant in the field - its focus on MR-TADF is relatively weak. The examples given address only a few examples - most of whom have limited practical use due to the rare elements included. The errors compared to experiments are not huge making the results impressive - but I would like to see a more comprehensive analysis of its performance across other examples.

The authors sincerely thank the reviewer for the comment which gives us a chance to comprehensively analyse the performance of our method across examples other than the MR-TADF emitters reported in the original manuscript. TADF emitters are categorised into fluorescent emitters having short-range charge-transfer (SRCT) excitations and long-range charge-transfer (LRCT) excitations. The MR-TADF emitters are categorised into SRCT-type molecules (Meng, G. et al., *Nat. Commun.* 14, 2394 (2023)). LRCT-type molecules are further divided into through-bond LRCT (TB-LRCT) and through-space LRCT (TS-LRCT). Di, D. et al. (*Science* 356, 159-163 (2017)) and Hamze, R. et al. (*Science* 363, 601 (2019)) reported another type of TADF emitters containing Au and Cu.

To demonstrate the robustness of our method, we carried out the calculations also for all these different type emitters, according to the reviewer's suggestion: 1) TB-LRCT type molecules with different connections and different numbers of donors and acceptors (*m*-3CzIPN, *o*-3CzIPN, 4CzIPN, 5CzBN), 2) TB-LRCT type molecules

with different types of donors and acceptors (PIC-TRZ2, DMAC-DPS), 3) a TB-LRCT type molecule containing a metal (CMA2), and 4) a TS-LRCT type molecule (TpAT-tFFO).

The reasons why we have chosen these molecules are as follows:

- i) *m*-3CzIPN, *o*-3CzIPN, 4CzIPN, and 5CzBN are carbazole-cyano derivatives showing high luminescence efficiency. They are composed only of C, H, and N atoms. Among the carbazole-cyano derivatives reported to date, 4CzIPN (Uoyama, H. et al., *Nature* 492, 234-238 (2012)) is the most well-known organic TADF emitter that triggered a paradigm shift from phosphorescence to TADF in organic light-emitting diodes (OLEDs). We investigated the effects of different bonding positions and different numbers of donors and acceptors.
- ii) PIC-TRZ2 (Sato, K. et al., *Phys. Rev. Lett.* 110, 247401 (2013)) is the first nearly “zero-gap” TADF emitter having a quite small experimental S_1 - T_1 energy gap of ~ 20 meV. PIC-TRZ2 achieves a breakthrough in developing organic TADF emitters with weak exchange interaction and small S_1 - T_1 energy gap.
- iii) DMAC-DPS is the first efficient blue organic TADF emitter composed of highly twisted donor-acceptor units (Zhang, Q. et al., *Nat. Photon.* 8, 326-332 (2014)). DMAC-DPS has opened the way to molecular design of highly efficient blue TADF emitters.
- iv) CMA2 (Di, D. et al., *Science* 356, 159-163 (2017) and Hamze, R. et al., *Science* 363, 601 (2019)) is an organometallic TADF emitter composed of a linear donor-Cu bridge-acceptor structure, which is different from conventional donor-acceptor type TADF emitters in which donor and acceptor moieties are directly bonded. Unlike conventional phosphorescent emitters including Ir or Pt, CMA2 is composed of earth-abundant Cu. The Cu bridge realises moderate HOMO-LUMO spatial overlap and consequently, CMA2 shows both fast radiative decay and fast RISC, leading to efficient blue-emitting OLEDs.
- v) TpAT-tFFO (Wada, Y. et al., *Nat. Photon.* 14, 643-649 (2020)) is a TADF emitter having TS-LRCT type excitations. Despite the absence of heavy atoms, TpAT-tFFO realises a very large RISC rate constant exceeding 10^7 s⁻¹, which is far greater than those of conventional TADF emitters.

Calculated rate constants and PLQYs are listed in the following table. The experimental data are shown in the parentheses (blanks were not reported). Overall, the calculated rate constants and PLQYs well reproduced the experimental values, indicating that our method is also valid for all these types of molecules. In addition, our method clearly reveals that ISC and RISC mechanisms depend on molecules. *m*-3CzIPN and *o*-3CzIPN are good examples (they are different only in the position of one of the carbazoles). For *m*-3CzIPN, the ISC and RISC proceed predominantly via T_2 state ($k_{\text{toISC}}(S_1 \rightarrow T_2)$ (2.2×10^8 s⁻¹) is one order of magnitude greater than $k_{\text{toISC}}(S_1 \rightarrow T_1)$ (0.2×10^8 s⁻¹); $k_{\text{toRISC}}(T_2 \rightarrow S_1)$ (1.1×10^5 s⁻¹) is also greater than $k_{\text{toRISC}}(T_1 \rightarrow S_1)$ (0.1×10^5 s⁻¹)). In sharp contrast, for *o*-3CzIPN, the ISC and RISC proceed predominantly via T_1 state ($k_{\text{toISC}}(S_1 \rightarrow T_2)$ (0.1×10^7 s⁻¹) < $k_{\text{toISC}}(S_1 \rightarrow T_1)$ (2.0×10^7 s⁻¹); $k_{\text{toRISC}}(T_2 \rightarrow S_1)$ (0.1×10^5 s⁻¹) < $k_{\text{toRISC}}(T_1 \rightarrow S_1)$ (6.7×10^5 s⁻¹)). For 4CzIPN, the ISC and RISC occur via both the T_1 and T_2 mediated pathways.

	m -3CzIPN	o -3CzIPN	4CzIPN	5CzBN	PIC-TRZ2	DMAC-DPS	CMA2	TpAT-tFFO
k_{ISC}	2.4×10^8 (1.1×10^8) ^a	2.1×10^7 (6.0×10^7) ^a	11×10^7 (7.0×10^7) ^a	4.9×10^8 (2.5×10^8) ^a	1.3×10^7 (1.2×10^7) ^b	8.8×10^7 (3.7×10^7) ^c	1.9×10^9	19×10^7 (5.3×10^7) ^e
$k_{\text{ISC}}(\text{S}_1 \rightarrow \text{T}_1)$	0.2×10^8	2.0×10^7	4.0×10^7	0.3×10^8	1.3×10^7	8.8×10^7	1.9×10^9	1.0×10^7
$k_{\text{ISC}}(\text{S}_1 \rightarrow \text{T}_2)$	2.2×10^8	0.1×10^7	7.3×10^7	4.6×10^8	2.3×10^5			18×10^7
k_{RISC}	1.2×10^5 (2.7×10^5) ^a	6.8×10^5 (6.6×10^5) ^a	4.4×10^5 (8.8×10^5) ^a	2.9×10^5 (2.2×10^5) ^a	1.9×10^6 (3.5×10^6) ^b	5.6×10^5 (5.6×10^5) ^c	1.7×10^7	2.0×10^7 (1.2×10^7) ^e
$k_{\text{RISC}}(\text{T}_1 \rightarrow \text{S}_1)$	0.1×10^5	6.7×10^5	1.6×10^5	0.2×10^5	1.8×10^6	5.6×10^5	1.7×10^7	0.1×10^7
$k_{\text{RISC}}(\text{T}_2 \rightarrow \text{S}_1)$	1.1×10^5	0.1×10^5	2.8×10^5	2.7×10^5	0.1×10^6			1.9×10^7
$k_{\text{IOF}} (\text{s}^{-1})$	0.9×10^7 (1.5×10^7) ^a	1.0×10^7 (1.2×10^7) ^a	1.4×10^7 (1.8×10^7) ^a	4.1×10^7 (1.9×10^7) ^a	1.3×10^6 (4.9×10^5) ^b	1.4×10^7 (1.4×10^7) ^c	1.1×10^7	1.9×10^6 (1.1×10^6) ^e
$k_{\text{NR}} (\text{s}^{-1})$	1.7×10^6	2.4×10^6	2.8×10^6	4.3×10^6	9.6×10^5	2.3×10^6	4.9×10^6	3.1×10^5 (2.0×10^5) ^e
$k_{\text{TADF}} (\text{s}^{-1})$	0.4×10^4 (3.8×10^4) ^a	2.0×10^5 (1.3×10^5) ^a	0.5×10^5 (2.2×10^5) ^a	2.2×10^4 (2.1×10^4) ^a	1.4×10^5 (2.1×10^4) ^b	7.4×10^4 (2.1×10^4) ^c	1.0×10^5 (3.0×10^5) ^d	1.8×10^5 (2.3×10^5) ^e
Φ	0.84 (0.86) ^a	0.80 (0.86) ^a	0.83 (0.86) ^a	0.90 (0.75) ^a	0.57 (0.41) ^b	0.86 (0.86)	0.68 (0.68) ^d	0.86 (0.84) ^e

a) Measured in 10^{-5} M toluene solution (Noda, H. et al., *Nat. Mater.* 18, 1084-1090 (2019)).

b) Measured in a thin film of mCP (Sato, K. et al., *Phys. Rev. Lett.* 110, 247401 (2013); Aizawa, N., *Nat. Commun.* 11, 3909 (2020))

c) Measured in a thin film of DPEPO (Ahn, D. et al., *Org. Electron.* 59, 39 (2018); Aizawa, N., *Nat. Commun.* 11, 3909 (2020))

d) Measured in 10^{-5} M 2-MeTHF solution (Hamze, R. et al. *Science* 363, 601 (2019)).

e) Measured in 10^{-4} M toluene solution (Wada, Y. et al., *Nat. Photon.* 14, 643-649 (2020)).

In summary, our method performs well across examples other than the MR-TADF molecules, and we are confident that the results fully satisfy the request of the reviewer. A number of conventional (LRCT-type) TADF emitters have been developed since the seminal work of the Adachi research group (*Nature* 492, 234-238 (2012)). However, the conventional TADF emitters typically show broad emission spectra, which is a severe disadvantage in practical display applications. MR-TADF (SRCT) emitters (Hatakeyama, T. et al., *Adv. Mater.* 28, 2777-2781 (2016)) have emerged as the most promising materials for practical applications because of their narrow emission spectra. The interest of researchers in the field of OLEDs, hence most of recent reports, has already shifted from conventional TADFs to MR-TADFs. The only drawback of MR-TADF emitters is their slow RISC rate constants. On this background, in this study, we focus on the molecular design for accelerating RISC of MR-TADF. This is also effective to keep the word limit of *Nature Communications* and to clearly show the solution strategy of recent problem of MR-TADF. Here, we have simply modified the following original sentence “We have recently initiated application of this method to catalytic photooxygenation to inhibit aggregation of amyloid- β peptide as a therapeutic strategy for Alzheimer's disease.” to “We have also confirmed the applicability of this method to a wide range of TADF emitters other than MR-type molecules and to catalytic photooxygenation to inhibit aggregation of amyloid- β peptide as a therapeutic strategy for Alzheimer's disease.” (page 14, lines 2–4 in the revised manuscript). The above table will be published online as a supplementary "peer review file."

Response to the comments of reviewer #2

Reviewer's comment:

The manuscript by Shizu and Kaji presents a remarkable method for predicting rate constants of excited-state decay processes of MR-TADF emitters. The relevance of MR-TADF in OLEDs is obvious, but its performance problem due to slow RISC is a major concern. The authors address this issue by quantum chemical calculations to quantitatively predict the rate constants and resulting quantum yields for MR-TADF emitters. The versatility of their approach, with potential applications in various research areas, adds significant value to the study. I would recommend publication in Nature Communications after addressing the following concerns.

The authors sincerely thank the reviewer for the meticulous reading and for recognizing the importance of our manuscript. We have revised the original manuscript according to the reviewer's comments below. Important changes are highlighted by yellow in the revised manuscript.

Reviewer's comment 1:

1. The title "Quantitative prediction of rate constants and its application to organic emitters" may sound misleading, since the calculated values are the rate constants of excited-state decays in a single molecule. The same is true of their claim in the abstract.

The authors appreciate the reviewer's comment. We searched for the article titles of theoretical and computational studies dealing with single-molecule organic emitter systems and found that in many cases, "*organic emitters*" or "*fluorescence emitters*" were used in the titles, even though they treated single-molecule systems. The followings are only some examples:

- i) "On predicting the excited-state properties of thermally activated delayed *fluorescence emitters*"
Penfold, T. J.
J. Phys. Chem. C 119, 13535-13544 (2015).
- ii) "Theoretical investigation of the singlet–triplet splittings for carbazole-based thermally activated delayed *fluorescence emitters*"
Liang, K. and Zhang, X. et al.
Phys. Chem. Chem. Phys. 18, 26623-26629 (2016).
- iii) "Up-conversion intersystem crossing rates in *organic emitters* for thermally activated delayed fluorescence: Impact of the nature of singlet vs triplet excited states."
Samanta, P. K., Kim, D., Coropceanu, V. & Brédas, J.-L.
J. Am. Chem. Soc. 139, 4042-4051 (2017).
- iv) "Purely *organic emitters* for multiresonant thermally activated delay fluorescence: design of highly efficient sulfur and selenium derivatives"
Pratik, S. M., Coropceanu, V. & Brédas, J.-L.
ACS Mater. Lett. 4, 440-447 (2022).
- v) Shi, Y. and Peng, Q. et al.

“Optimal dihedral angle in twisted donor–acceptor organic emitters for maximised thermally activated delayed fluorescence”

Angew. Chem. Int. Ed. 61, e202213463 (2022).

All these articles have carried out calculations for single molecules. The same is true for experimental publications; plurals have been used for dilute solution systems. In our manuscript, we have carried out calculations for several molecules. Therefore, we believe that plural form reflects the actual contents of our study. Accordingly, we retained “organic emitters” in the title of the revised manuscript. On the other hand, the reviewer’s comment is also reasonable. To make the content clear and to reflect the reviewers’ comment, we include the word, single molecule, in the abstract.

The original version:

“We first showed a quantum chemical calculation method for quantitatively reproducing all experimentally obtained rate constants and quantum yields for previously synthesised MR–TADF emitters.”

The revised version (page 2, line 10 in the abstract section):

“We first showed a *single-molecule* quantum chemical calculation method for quantitatively reproducing all experimentally obtained rate constants and quantum yields for previously synthesised MR–TADF emitters.”

Reviewer’s comment 2:

2. While this manuscript delves into the various practices of MR-TADF emitters, a clearer presentation of the theoretical advances and the reason why they enable the quantitative predictions would greatly aid the reader's understanding.

Thank you for the valuable suggestion which gives us a chance to prepare a better manuscript. Understanding the excited-state decay mechanism of MR-TADF emitters is important for designing novel materials with enhanced TADF properties. To understand the decay mechanism of a TADF emitter, it is common to determine k_F , k_{IC} , k_{ISC} , and k_{RISC} by combining experimental transient photoluminescence decay curve fitting and PLQY. A comprehensive understanding of the emission mechanism is achieved only when the rate constants for all elementary electronic transitions have been determined. However, it is difficult to determine all rate constants when the number of the rate constants is larger than that of experimentally determined fitting parameters (specifically, when singlet and triplet states energetically higher than S_1 and T_1 are involved). Our theoretical method allows us to quantitatively predict rate constants, including those inaccessible from experiments, providing comprehensive understanding of the TADF mechanism. Therefore, our method proposed in this study offers a guideline for designing TADF emitters with enhanced properties.

In the revised manuscript, we added a related description in the revised manuscript (page 5, lines 1–13).

Reviewer’s comment 3:

3. The manuscript does not specify the method used to calculate the SOCs, vibronic coupling constants, and T1-T2 transition dipole moments (it just says that it was done by their own method). Providing this information is essential for the reproducibility of their study.

The authors appreciate the reviewer's comment. Calculations of the SOCs, vibronic coupling constants, and T_1 - T_2 transition dipole moments were based on the method proposed by McMurchie and Davidson (McMurchie, L. E. & Davidson, E. R. One- and two-electron integrals over cartesian gaussian functions. *J. Comput. Phys.* 26, 218-231 (1978)). We added descriptions regarding the method of calculating SOCs, vibronic coupling constants, and T_1 - T_2 transition dipole moments in the revised version of Supplementary Method 2. We also described it in the Method section of the revised manuscript: “the SOCs, vibronic coupling constants, and T_1 - T_2 transition dipole moments were calculated with the method proposed by McMurchie and Davidson (Supplementary Method 2)” (page 24, lines 2–4 from the bottom in the revised manuscript).

Reviewer's comment 4:

4. The authors deliberately employed the TDA-B2-PLYP/def2-TZVP to calculate ΔE_{ST} and the TD-B3LYP/6-31G(d)+SDD for the other parameters in predicting the rate constants. The manuscript could benefit from a justification for mixing different methods to ensure that the accuracy of their predictions is not primarily due to error compensation (error canceling).

Thank you for the valuable comment. For MR-TADF emitters, because it is difficult to predict all relevant molecular properties with a single density functional or wave function method, it has become common to use different methods to different molecular properties. Recent theoretical studies regarding MR-TADF are listed in the following table. As shown in the table, different methods are used for calculating S_1 - T_1 energy difference (ΔE_{ST}) and the other parameters (vibronic coupling (VC), transition dipole moment (TDM), and spin-orbit coupling (SOC)).

Conventional TD–DFT methods have been found to substantially overestimate ΔE_{STS} of MR-TADF emitters, which is solved by calculations including double-excitation configurations (Pershin, A. et al. *Nat. Commun.* 10, 597 (2019)). Hence, it has become standard to calculate ΔE_{STS} of MR-TADF emitters using the ADC(2) or SCS-CC2 method (because ADC(2) and SCS-CC2 involve double-excitation configurations). Tamm–Dancoff approximation (TDA)–DFT methods with double hybrid density functionals such as TDA–B2-PLYP are emerging alternative approaches for considering double-excitation configurations (Sancho-García, J. C. et al. *J. Chem. Phys.* 156, 034105 (2021)). TDA–DFT methods with double hybrid density functionals have the advantage of low computational cost compared to ADC(2) and SCS–CC2. Meanwhile, for the other parameters, conventional TD–DFT or TDA-DFT methods are still used because they provide reliable results at low computational costs. On this background, we calculated ΔE_{ST} with the TDA-B2-PLYP method and the other parameters with the B3LYP method (page 7, lines 8–12 from the bottom in the revised manuscript).

	Geometry optimisation	ΔE_{ST}	VC/TDM/SOC
Tanaka, H. and Hatakeyama, T. et al. Angew. Chem. Int. Ed. , 60, 17910-17914 (2021)	B3LYP/ 6-311+G(d)	ADC(2)/ def2-SVP	PBE0/ DZP
Kim, I. and Kim, D-S. et al. JACS Au , 1, 987-997 (2021)	SCS-ADC(2)/ def2-SVP	SCS-CC2/ def2-SVP	ω^* B97X ^a / def2-SVP
Lin, S. and Shuai, Z. et al. ACS Mater. Lett. , 4, 487-496 (2022)	B3LYP/ 6-31G(d)	SCS-CC2/ def2-TZVP	B3LYP/ 6-31G(d)

a) ω^* means that the ω value is optimised.

Response to the comments of reviewer #3

Reviewer's comment:

In this manuscript, Kaji and coworkers theoretically predicted the rate constants of each process involved in the multi-resonance thermally activated delayed fluorescence (MR-TADF) of a series of MR-TADF emitters consisting of different heavy atoms by quantum chemical calculations. They comprehensively studied the effects of heavy atoms on the kinetics of MR-TADF and the photophysics of the emitters. They showed a clear picture on the kinetics of each elementary process, and the contribution to the total rate constants by the S1, T1 and T2 states was analysed extensively. However, there are numerous studies involving the calculations of these elementary processes, for example, the rate constant calculations using MOMAP by Zhigang Shuai's research group, and it is predictable that the emitter involving more heavy polonium may have phosphorescence even without calculations. Therefore, together with the following comments, this work is not novel enough to be published on Nat. Comm.

The authors sincerely thank the reviewer for the valuable comment which gives us a chance to prepare a better manuscript. We have revised the original manuscript according to the reviewer's comments below. Important changes are highlighted by yellow in the revised manuscript.

Reviewer's comment 1:

1. In this work, the authors have not discussed the influences of heavy atoms on the stability of the MR-TADF emitters, which is critical to the lifetime of the devices. In the devices, there will be some host-guest interactions between the emitter and the host, and so the MR-TADF compounds may undergo dissociation to form radicals. In addition, the poor operation lifetime of OLED was confirmed to arise from the weak bond energy between the carbon and heavy atoms (Adv. Funct. Mater. 2023, 2306880). The strategy of introducing heavy atoms to the MR-TADF molecules is meaningless if the stability of the MR-TADF emitters is low, even the RISC is accelerated without sacrificing radiative decay and PLQY.

Our target of this study is to first establish quantitative predictions of all rate constants and quantum yields in OLED emitters. The results have been successful not only for MR-TADF but also other emitters as we responded to the comment 2 of reviewer #1. Regarding the prediction of device lifetime, it is difficult at this stage because the number of reports on experimental lifetimes is limited and also they are highly group-dependent. Moreover, it is affected not only by emitters but also by other various factors such as charge recombination regions, impurities in the device, and degradation of surrounding materials. For conventional D-A type TADF molecules and MR-TADF with donor segment, analysis by bond dissociation energy (BDE) has been often carried out. In these cases, dissociation is assumed to occur between segments connected by a single bond. In contrast, there are no such single bonds and all atoms are connected by two bonds for the molecules in this study. Therefore, it is uncertain whether the same BDE analysis is valid for the molecules in this study. Recent discussion with Samsung group (who are experts on device lifetime) indicated that BDE is not directly related to device lifetime. For these reasons, we would like to leave device lifetime outside the scope of this study.

Reviewer's comment 2:

2. On page 5, line 86, the authors quote the experimental k_{RISC} values from Yang's work (ref. 22), which are 4.3×10^4 and $1.9 \times 10^5 \text{ s}^{-1}$, but in ref. 22, the k_{RISC} values provided in Table 1 are not consistent with them.

Thank you for the careful reading. Experimental photophysical properties for BNOO and BNSS have been reported in two papers both by Chuluo Yang's group (ref. 10 (*Nat. Photon.* 16, 803–810 (2022)) and ref. 22 (*Chem. Eng. J.* 426, 131169 (2021))). They have reported three different k_{toRISC} values; for a system 1 wt% doped in DMIC-TRZ host (ref. 10), a 10^{-5} M toluene solution system, and a system 1 wt% doped in mCBP:PO-T2T mixed-host (ref. 22). We have not cited values in ref. 22. Instead, we cite the experimental values in ref. 10, because the experimental condition is the same for all emitters in ref. 10 and both k_{toRISC} and k_{toISC} values are reported (no k_{toISC} values are reported in ref. 22). The sentence in the original manuscript, “*We first investigated the RISC mechanism of previously synthesised MR-TADF emitters (BNOO,²² BNSS,²² and BNSeSe¹⁰) (Fig. 1), reported by Yang's group.*”, is misleading as the reviewer pointed out; the sentence gives the impression that the experimental k_{RISC} values for BNOO and BNSS are from ref. 22 and that for BNSeSe is from ref. 10. To avoid the misunderstanding, in the revised manuscript (page 5, lines 5–7 from the bottom), we rewrote it as “*We first investigated the RISC mechanism of previously synthesised MR-TADF emitters (BNOO, BNSS, and BNSeSe) (Fig. 1), reported by Yang's group¹⁰.*”.

Reviewer's comment 3:

3. On page 5, line 84, the calculated k_{RISC} of BNSS is 27 times of that of BNOO, but in ref. 22, it is around 2-3 times. How do the authors explain such great discrepancy? The statement of “indicating the validity of our calculation method” is thus inappropriate.

The calculated k_{toRISC} of $2.5 \times 10^5 \text{ s}^{-1}$ for BNSS agrees well with the experimental k_{toRISC} of $1.9 \times 10^5 \text{ s}^{-1}$. Therefore, the difference originates from the calculated k_{toRISC} for BNOO, $9.2 \times 10^3 \text{ s}^{-1}$ (1/27 of that for BNSS, $2.5 \times 10^5 \text{ s}^{-1}$). However, different rate constants have often been reported experimentally in different papers for the same emitter molecule, as exemplified by the reviewer's comment 2 above. More specifically, often, rate constants are derived by double exponential fitting of an experimental transient PL curve that cannot be described by a simple double exponential. The experimental curve also depends on experimental conditions. PLQY values also vary from paper to paper and depend on experimental conditions even for the same molecule. These mean that even the experimental values have certain degree of range. In this sense, it is fair to say that the calculated k_{toRISC} of $9.2 \times 10^3 \text{ s}^{-1}$ for BNOO in this study (nearly 10^4 s^{-1}) is in reasonable agreement with the experimental value of $4.3 \times 10^4 \text{ s}^{-1}$. Incidentally, the origin of the calculated k_{toRISC} of BNOO ($9.2 \times 10^3 \text{ s}^{-1}$) is an overestimation of $\Delta E(\text{T}_1 \rightarrow \text{S}_1)$ (the calculated $\Delta E(\text{T}_1 \rightarrow \text{S}_1)$ is 0.21 eV while the experimental value is 0.15 eV). When the experimental $\Delta E(\text{T}_1 \rightarrow \text{S}_1)$ is used, k_{toRISC} is calculated to be $8.2 \times 10^4 \text{ s}^{-1}$, much closer to the experimental value of $4.3 \times 10^4 \text{ s}^{-1}$.

Overall, the statement of “indicating the validity of our calculation method” is appropriate from the agreements between the calculated k_{toISC} (8.6×10^7 , 2.0×10^8 , and $1.0 \times 10^9 \text{ s}^{-1}$) and the experimental k_{toISC} (7.5×10^7 , 1.5×10^8 , and $4.9 \times 10^8 \text{ s}^{-1}$) and those of the calculated k_{toRISC} (9.2×10^3 , 2.5×10^5 , and $1.5 \times 10^6 \text{ s}^{-1}$) and the experimental k_{toRISC} (4.3×10^4 , 1.9×10^5 , and $2.0 \times 10^6 \text{ s}^{-1}$) for BNOO, BNSS, and BNSeSe, respectively. The results for other emitters in the comment 2 of reviewer #1 also support the validity. We explain this point in page 6, lines 6–10 in the revised

manuscript.

Reviewer's comment 4:

4. On page 6, line 104, how to predict the FWHM of the PL spectra theoretically?

We theoretically calculated the FWHM values of the emission spectra by the vertical gradient method implemented in the ORCA 5.0.3 program package. The ORCA input file for calculating the emission spectra is described in Supplementary Information (Supplementary Method 5).

Reviewer's comment 5:

5. On page 6, line 109-126, it is better to include it in the section of computational details instead in discussion section. The same suggestion on lines 143-151 on page 7.

The authors sincerely thank the reviewer for the comment which helps us to prepare a more readable manuscript. To reflect the reviewer's comment, we moved the descriptions regarding the calculation method (page 6, line 109-126 and page 7, 143-151 in the original manuscript) to Methods section (page 24, line 4 – page 25, line 3).

Reviewer's comment 6:

6. On page 11, line 240, the authors state that "The large spatial overlap between HOMO and LUMO+1 around the C=O groups lowers the T₂ energy level, leading to the large k_{toRISC}." The lowering of the T₂ energy would increase the barrier for the T₂→S₁ transition. Why does it lead to large k_{toRISC}?

Thank you for the valuable question. Generally, the rate constant for the T₂→S₁ RISC decreases with lowering the T₂ energy, when T₂ is lower than S₁ in energy (this is the case of BNCOCO). In this sense, as the reviewer pointed out, lowering the T₂ energy has a disadvantage in terms of the single T₂→S₁ process. However, lowering the T₂ energy also decreases the T₁-T₂ energy gap, resulting in acceleration of the T₁→T₂ internal up-conversion (IC). For BNCOCO, the effect of accelerating the T₁→T₂ IC is greater than that of slowing down the T₂→S₁ RISC; the overall T₁→T₂→S₁ RISC rate constant increases by lowering the T₂ energy. We have already explained this point in page 11, lines 14–20 of the original manuscript.

Reviewer's comment 7:

7. In supplementary Figure 4, does the simulation of the calculated emission spectra involve the broadening by any functions like Gaussian, Lorentzian and Voigt functions?

As described in the caption of Supplementary Figure 4 in both the original and the revised manuscripts, we used the "Gaussian" function.

REVIEWER COMMENTS

Reviewer #2 (Remarks to the Author):

The authors have made an important effort to address the questions raised. Overall, their response is largely satisfactory; however, I would like to suggest that the authors further refine some of their points:

1. Specifics on the theoretical framework newly developed and the mechanisms enabling quantitative predictions should be explained, as also requested in my previous question: “2. While this manuscript delves into the various practices of MR-TADF emitters, a clearer presentation of the theoretical advances and the reason why they enable the quantitative predictions would greatly aid the reader's understanding”.

2. The McMurchie-Davidson formulation (1978) offers a general method for calculating two-electron integrals. Beyond merely specifying the use of this formulation, I suggest that the authors provide the actual codes employed in their calculations. Including these codes will not only improve reproducibility but also ease validation efforts by fellow researchers. Furthermore, a detailed documentation of the codes would benefit future research across various fields.

Upon addressing the above concerns, I am confident that this work will make a valuable contribution to Nature Communications.

Reviewer #3 (Remarks to the Author):

The authors have addressed most of the raised issues with detailed clarifications and elaborations. They have conducted a systematic theoretical study and investigated the impact of heavy atom effect on the RISC process. They provide a “theoretically” good guideline for researchers to consider when designing MRTADF materials. Nevertheless, it is noteworthy that if these guidelines have actual applications on future OLED research. For example, they revealed that $\Delta E(T1-T2)$ is also crucial in the T2-mediated RISC process. The suggestion of minimizing $\Delta E(T1-T2)$ is a bit vague. Instead, to make the manuscript better, they should tell the readers how to actually control the $\Delta E(T1-T2)$. The majority of the discussion in this manuscript is on the theoretical level, therefore, it would be more suitable to be published on some computational or theoretical chemistry journals like “J. Chem. Theory Comput.” and “J. Comp. Chem.”

Response to the comments of reviewer #2

Reviewer's comment:

The authors have made an important effort to address the questions raised. Overall, their response is largely satisfactory; however, I would like to suggest that the authors further refine some of their points:

The authors sincerely thank the reviewer for the valuable comments. We have further revised the manuscript according to the reviewer's comments below. Important changes are highlighted by yellow in the revised manuscript.

Reviewer's comment 1:

1. Specifics on the theoretical framework newly developed and the mechanisms enabling quantitative predictions should be explained, as also requested in my previous question: "2. While this manuscript delves into the various practices of MR-TADF emitters, a clearer presentation of the theoretical advances and the reason why they enable the quantitative predictions would greatly aid the reader's understanding".

In developing the method to quantitatively predict all rate constants, the most problematic point was the quantitative evaluation of the multiple routes that ISC and RISC consist of. We first proposed a theoretical method of predicting the energy level alignments including higher lying states as well as S_1 and T_1 by combining conventional B3LYP and B2-PLYP double-hybrid functionals. We solve the above problem by devising a method of calculating RISC (and ISC) rate constants considering RISC (and ISC) mechanisms consisting of multiple pathways. Although only S_1 , T_1 , and T_2 are involved in the emission mechanism of molecules in this study, the method proposed here is robust enough to be applied to the case where many more higher energy states are involved. We added this point in the revised manuscript (Lines 315 – 323).

Reviewer's comment 2:

2. The McMurchie-Davidson formulation (1978) offers a general method for calculating two-electron integrals. Beyond merely specifying the use of this formulation, I suggest that the authors provide the actual codes employed in their calculations. Including these codes will not only improve reproducibility but also ease validation efforts by fellow researchers. Furthermore, a detailed documentation of the codes would be benefit future research across various fields.

Thank you for the valuable suggestion. The Fortran90 source files and Makefile for computing vibronic couplings and spin-orbit couplings with the McMurchie-Davidson formulation are freely available on a GitHub repository (<https://github.com/KatsuyukiShizu/d77.git>) under the GNU General Public License, version 3. Sample files for calculating vibronic couplings and spin-orbit couplings are also available. We believe that providing the source code contributes significantly to the future research across various fields. In the revised manuscript, we added the URL of the GitHub repository in the Code availability statement (Page 21). We also added the DOI link generated by figshare in the Data availability statement (Page 21), where the Gaussian 16 and ORCA input and output files are deposited.

Response to the comments of reviewer #3

Reviewer's comment:

The authors have addressed most of the raised issues with detailed clarifications and elaborations. They have conducted a systematic theoretical study and investigated the impact of heavy atom effect on the RISC process. They provide a “theoretically” good guideline for researchers to consider when designing MRTADF materials. Nevertheless, it is noteworthy that if these guidelines have actual applications on future OLED research. For example, they revealed that $\Delta E(T_1-T_2)$ is also crucial in the T₂-mediated RISC process. The suggestion of minimizing $\Delta E(T_1-T_2)$ is a bit vague. Instead, to make the manuscript better, they should tell the readers how to actually control the $\Delta E(T_1-T_2)$. The majority of the discussion in this manuscript is on the theoretical level, therefore, it would be more suitable to be published on some computational or theoretical chemistry journals like “J. Chem. Theory Comput.” and “J. Comp. Chem.”

The authors sincerely thank the reviewer for providing the opportunity to improve the manuscript. We have revised the manuscript according to the reviewer's comment. Important changes are highlighted by yellow in the revised manuscript.

We first consider the case where T₁ and T₂ consist of HOMO → LUMO (H → L) and HOMO-1 → LUMO (H-1 → L) transitions, respectively. The Slater determinants describing T₁ and T₂ are represented as follows:

The T₁ and T₂ energies can be expressed as (Ref 44, Szabo, A. & Ostlund, N. S. Modern quantum chemistry: Introduction to advanced electronic structure theory pp. 87-89, Dover Publications, New York, 1996)

$$E(T_1) = 2h_{H-1H-1} + h_{HH} + h_{LL} + J_{H-1H-1} + 2J_{H-1H} + 2J_{H-1L} + J_{HL} - K_{H-1H} - K_{HL}$$

$$E(T_2) = h_{H-1H-1} + 2h_{HH} + h_{LL} + J_{HH} + 2J_{H-1H} + J_{H-1L} + 2J_{HL} - K_{H-1H} - K_{HL}$$

where h denotes the core integral (kinetic and potential energies), J denotes the Coulomb integral, and K denotes the exchange integral. Then, $\Delta E(T_2 - T_1)$ is written as

$$\Delta E(T_2 - T_1) = (h_{HH} - h_{H-1H-1}) + (J_{HH} - J_{H-1H-1}) + (J_{HL} - J_{H-1L})$$

A simple approach of minimizing $\Delta E(T_2 - T_1)$ is to decrease the first term ($h_{HH} - h_{H-1H-1}$) by expanding the HOMO and HOMO-1 distributions (the second and third terms of $\Delta E(T_2 - T_1)$, expressed in terms of the Coulomb integrals, are not easy to control).

We next consider the case where T₁ and T₂ consist of the HOMO → LUMO (H → L) and HOMO → LUMO+1 (H → L+1) transitions, respectively. In this case, the Slater determinants describing T₁ and T₂ are represented as follows:

The T_1 and T_2 energies can be expressed as

$$E(T_1) = h_{HH} + h_{LL} + J_{HL} - K_{HL}$$

$$E(T_2) = h_{HH} + h_{L+1L+1} + J_{HL+1} - K_{HL+1}$$

Hence, $\Delta E(T_2 - T_1)$ is written as

$$\Delta E(T_2 - T_1) = (h_{L+1L+1} - h_{LL}) + (J_{HL+1} - J_{HL}) - (K_{HL+1} - K_{HL})$$

A simple approach of minimizing $\Delta E(T_2 - T_1)$ is to decrease the first term ($h_{L+1L+1} - h_{LL}$) by expanding the LUMO+1 and LUMO distributions (the second and third terms of $\Delta E(T_2 - T_1)$ are not easy to control).

Regardless of whether T_2 is described as HOMO-1 \rightarrow LUMO or HOMO \rightarrow LUMO+1 transition, expanding molecular orbitals relevant for T_1 and T_2 is a simple way to decrease $\Delta E(T_2 - T_1)$ and accelerates the T_2 -mediated RISC process. Comparison of v-DABNA-core and V-DABNA-core (Ref 44, Oda, S. *et al. J. Am. Chem. Soc.* **144**, 106 (2022)) is a good example. V-DABNA-core has a larger π - conjugation than v-DABNA-core and hence, V-DABNA-core shows a smaller $\Delta E(T_2 - T_1)$ of 98 meV than v-DABNA-core (147 meV).

We added the above description in the revised manuscript (Lines 245 – 263; Refs 44 and 45) and in Supplementary Information (page S33, Supplementary Method 6 and Supplementary Figure 8; page S34, Ref 11).

Although this study is mainly theoretical, we believe that our findings will be of interest to a wide range of materials scientists, chemists, and physicists, including experimental researchers.

REVIEWERS' COMMENTS

Reviewer #2 (Remarks to the Author):

All of my comments have been addressed and answered convincingly; therefore, I can now recommend the publication of the revised manuscript.

Reviewer #2 (Remarks on code availability):

I was able to install and run the code using the example input files provided in GitHub.

Reviewer #3 (Remarks to the Author):

The authors have addressed the issue regarding the reduction of the $\Delta E(T2-T1)$, with detailed explanations and suitable equations provided. Two possible cases have been considered in the discussion, and they conclude that one of the ways for reducing $\Delta E(T2-T1)$ is expanding the molecular orbitals for T1 and T2 states. Instead of the example provided by the authors, I am interested to know if there are any other examples to support the generalization.

Response to the comments of reviewer #2

Reviewer's comment (Remarks to the Author):

All of my comments have been addressed and answered convincingly; therefore, I can now recommend the publication of the revised manuscript.

Reviewer's comment (Remarks on code availability):

I was able to install and run the code using the example input files provided in GitHub.

The authors sincerely thank the reviewer for recognizing our effort to address the reviewer's questions. We also appreciate the reviewer taking the time out of your busy schedule to confirm that our code works without any problems and to provide positive comments on our revised manuscript.

Response to the comments of reviewer #3

Reviewer's comment:

The authors have addressed the issue regarding the reduction of the $\Delta E(T_2-T_1)$, with detailed explanations and suitable equations provided. Two possible cases have been considered in the discussion, and they conclude that one of the ways for reducing $\Delta E(T_2-T_1)$ is expanding the molecular orbitals for T1 and T2 states. Instead of the example provided by the authors, I am interested to know if there are any other examples to support the generalization.

The authors sincerely thank the reviewer for the valuable comment. We have further revised the manuscript in response to the reviewer's comment. Important changes are highlighted by yellow in the revised manuscript.

Expanding molecular orbitals relevant to T_1 and T_2 is a simple approach to reduce $\Delta E(T_2 - T_1)$ which accelerates the T_2 -mediated RISC process. In the revised manuscript, we cited additional five examples (Refs 46-50) that support the generalization of this approach. In the following, we summarized $\Delta E(T_2 - T_1)$ values in Refs 45-50. The same table was added as Supplementary Table 15 in the revised Supplementary Information. We are deeply grateful for your valuable comment, which has allowed us to further improve this study.

MR-TADF molecules with $\Delta E(T_2-T_1)$		References (in the main text)
		12 (45)
0.147 eV	>	0.098 eV
		13 (46)
0.28 eV	>	0.12 eV
		14 (47)
0.46 eV	>	0.33 eV
		15 (48)
0.21 eV	>	0.07 eV
		16 (49)
0.58 eV	>	0.28 eV
		17 (50)
0.81 eV	>	0.04 eV